# Online Bandit Learning with Offline Preference Data

## Abstract

Reinforcement Learning with Human Feedback (RLHF) is at the core of fine-tuning methods for generative AI models for language and images. Such feedback is often sought as preference feedback from human raters, as opposed to eliciting scores since the latter tends to be noisy. On the other hand, RL theory and algorithms predominantly assume that a reward feedback is available. In particular, approaches for online learning that can be helpful in adaptive data collection via active learning cannot incorporate offline preference data. In this paper, we adopt a finite-armed linear bandit model as a prototypical model of online learning. We consider an offline preference dataset to be available generated by an rater of unknown 'competence'. We propose warmPref − PS, a posterior sampling algorithm for online learning that can be warm-started with an offline dataset with noisy preference feedback. We show that by modeling the 'competence' of the rater that generated it, we are able to use such a dataset most effectively. We support our claims with novel theoretical analysis of its Bayesian regret, as well as, extensive empirical evaluation of an approximate loss function that optimizes for infinitely many arms, and performs substantially better than baselines.

## 1 Introduction

In the development of generative AI models for language and image generation, it has proven quite effective to first 'pretrain' with a very large *offline* dataset followed by *online* reinforcement learning (RL)-based 'fine-tuning' with small amounts of high quality Human Feedback (HF) data to improve alignment with human preferences. Although preference based HF data are less noisy and easier to aggregate over multiple raters, absolute score based HF data are generally more informative than relative preferences, and designing mechanisms that find optimal tradeoffs between these different feedback modalities is critical to scaling RLHF.

In practice today, there is already a lot of *offline* preference data available to the models. These preference data are generated from batches sent to human annotators to provide preferences on. However, for task specific online finetuning, reward models (called 'AutoRaters' (Anil et al., 2023)) are used for active learning. This is because it is expensive to do active learning with human raters in an online manner. The reward models are typically trained on these preference datasets in an offline manner, and are used to provide reward feedback in the online phase (Achiam et al., 2023; Anil et al., 2023). The setting of offline preferences and online numerical rewards is also applicable to the case where a foundational model aligned to general human preferences from an initial *offline* dataset needs to be rapidly personalized to the idiosyncratic *online* preferences of a particular user. We approach the challenge of minimizing *this* online learning. While it is trivial that collecting additional data with online finetuning will improve performance, *how* to effectively combine preference and numerical reward learning is highly nontrivial. Hence, in this setting where the online ratings are absolute scores, we propose a simple Bayesian algorithm for online learning that incorporates learning from an offline preference dataset. We note here that our problem formulation below is motivated by the practical relevance discussed above, and, to the best of our knowledge, no other work that formalizes and analyses this setting exists.

To formalize the practical relevance, we adopt a finite-armed linear bandit model, with arms corresponding to different generated model outputs, with indicated rater preferences available offline before starting the online phase when absolute reward scores from a user become available. To efficiently learn the optimal arm

selection strategy, we propose warmPref − PS, a posterior sampling-based Bayesian algorithm that naturally incorporates offline preference data and online reward feedback, and minimizes Bayesian regret.

**Relevance to RLHF.** Since RLHF can be modeled as a bandit problem with context-output pairs (Gheshlaghi Azar et al., 2024), our problem setting sits at the intersection of offline preference learning and online reward-based fine-tuning in RLHF. Tajwar et al. (2024) shows that when the reward optimum lies in low-probability regions of the reference policy $\pi_{\text{ref}}$ (i.e. KL divergence between optimal policy $\pi^\star$ and $\pi_{\text{ref}}$ is large) , on-policy sampling during online fine-tuning becomes crucial. This insight translates directly to our bandit setting, where maintaining a posterior over the reward function and updating via online sampling yields significantly improved performance compared to purely offline approaches, as seen in Figure 1. Further, Zhang et al. (2024) introduces a data relabeling scheme that augments offline binary preferences with explicit numerical reward values, avoiding "unlearning" of rejected yet high-quality outputs. This boosts generalization by better utilizing the full response space, a principle shown in our reward-based online phase, which outperforms standard DPO (Rafailov et al., 2024)

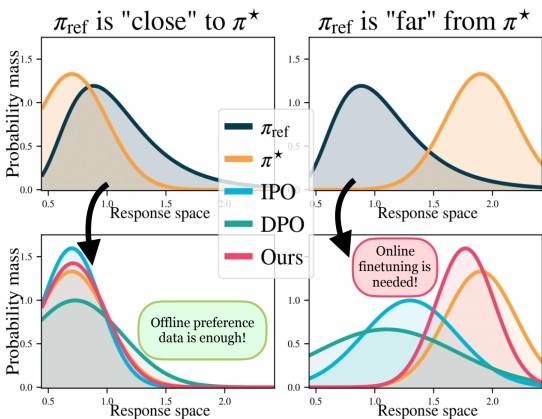

Figure 1: warmPref − PS (ours) comparison with IPO (Gheshlaghi Azar et al., 2024) and DPO (Rafailov et al., 2024). When divergence between $\pi_{\text{ref}}$ and $\pi^\star$ is large, on-policy online learning is needed for best results.

on multiple benchmarks. Tu et al. (2024); Bai et al. (2025) address the scarcity of real-time human feedback in online phases by using LLMs to generate self-augmented preferences. While we take a different route (i.e. direct reward-based online updates), both approaches highlight the same need: augmenting offline preference datasets with online learning.

**Contributions of this paper.** (i) We present the first online learning algorithm that incorporates *offline preference data* into online learning, even when it comes from a subpar expert. The key is that the algorithm is able to model and learn how 'competent' the expert is (with respect to the optimal policy). (ii) While our proposed algorithm is a natural extension of posterior sampling, it requires significantly different theoretical analysis due to the offline preference data (see Lemma 4.5 and Theorem 4.6). We provide novel theoretical guarantees on the minimum size of the offline dataset needed for it to allow learning of the optimal action. We also provide an upper bound on the algorithm's Bayesian regret that reveals the dependence of the offline dataset size, and the expert's 'competence'. (iii) We propose a practical version of our warmPref − PS algorithm , called Bootstrapped warmPref − PS, that is computationally tractable for an infinitely armed bandit environment, and establish its superior empirical performance with regard to baselines.

**Related work.** There is substantial literature on online learning for bandit models in various settings - finite-armed or linear, stochastic or adversarial, non-contextual or contextual models, etc. (Lattimore & Szepesvári, 2020). There is recent literature on utilizing offline data to improve learning regret during the online phase but these approaches either do not incorporate offline preference data (Shivaswamy & Joachims, 2012; Bouneffouf et al., 2019; Zhang et al., 2019; Banerjee et al., 2022; Agrawal et al., 2023), or solve the best arm identification problem which focuses on pure exploration (Agrawal et al., 2023). Furthermore, the quality of the offline data is not accounted for, which usually results in only a marginal regret reduction even while warm-starting with offline data. Ranking, comparison or preference feedback is considered in dueling bandit models (Dudík et al., 2015; Wu & Liu, 2016; Yan et al., 2022; Bengs et al., 2021; Szörényi et al., 2015; Agnihotri et al., 2023; 2019; Hu et al., 2022) but it is akin to active learning from preferences (Ailon, 2011) without incorporating a *given, fixed* offline preference dataset. Another set of works for the contextual bandit setting exist (Saha, 2021; Bengs et al., 2022; Li et al., 2024), however they cannot combine learning from rewards *and* preferences. The importance of offline dataset quality in imitation learning was first investigated in Beliaev et al. (2022b). Later, Hao et al. (2023) introduced an algorithm which leveraged offline reward feedback to warm start the online phase. While the algorithm uses offline reward feedback data, it cannot incorporate preference feedback as we do in this paper. Since, incorporation of preference

feedback is nontrivial, the regret analysis techniques (presented in Appendix A) are entirely different than for the case when reward feedback is available. To our best knowledge, ours is the first online bandit learning algorithm that can incorporate offline preference data.

## 2 Preliminaries

We model unknown quantities as random variables defined on a common probability space $(\Omega, \mathcal{F}, \mathbb{P})$. Now, consider a stochastic $K$-armed linear bandit problem with a set of actions, $\mathcal{A} = \{a_0, \ldots, a_K\} \subseteq \mathbb{R}^d$. The environment is characterized by a random vector $\theta \in \mathbb{R}^d$, with a prior distribution $\nu_0$. At time step $t$, the agent chooses an action $A_t \in \mathcal{A}$ and receives a reward $R_t$:

$$R_t = \langle A_t, \theta \rangle + \eta_t,$$

where $\eta_t \sim \mathcal{N}\left(0, \sigma^2\right)$ are i.i.d. sampled at each time step. For RLHF applications, the rewards might correspond to absolute score feedback by an individual rater on outputs (i.e., actions) generated by a foundational model. The agent's objective is to maximize $\sum_{t=1}^{T} \mathbb{E}[R_t]$, the expected total reward over horizon $T$, where the expectation is over the algorithm's decisions and the randomness in the environment. In addition, we also have an initial *offline preference* dataset $\mathcal{D}_0$, which is generated by human raters *with limited competence*. This offline dataset is a sequence of tuples of the form $\mathcal{D}_0 = \left(\left(\overline{A}_n^{(0)}, \overline{A}_n^{(1)}, Y_n\right)\right)_{n \in [N]}$, where $[N] := [1, 2, \ldots, N]$, $\overline{A}_n^{(0)}, \overline{A}_n^{(1)} \in \mathcal{A}$ are two actions, and $Y_n \in \{0, 1\}$ indicates the rater's preference. In particular, $Y_n = 0$ if the rater prefers action $\overline{A}_n^{(0)}$ to $\overline{A}_n^{(1)}$, and $Y_n = 1$ otherwise. In addition to the dataset size $N$, we characterize the offline dataset by: (i) an action sampling distribution $\mu$, where $\overline{A}_n^{(0)}$ and $\overline{A}_n^{(1)}$ are i.i.d. sampled from $\mu$; and (ii) assuming that given $\overline{A}^{(0)}$ and $\overline{A}^{(1)}$, the rater follows a *noisy* Bradley-Terry model (Bradley & Terry, 1952) and chooses $Y = 0$ (i.e., ranks action $\overline{A}_n^{(0)}$ above $\overline{A}_n^{(1)}$) with probability

$$P\left(Y = 0 \mid \overline{A}^{(0)}, \overline{A}^{(1)} ; \vartheta\right) = \frac{\exp\left(\beta \left\langle \overline{A}^{(0)}, \vartheta \right\rangle\right)}{\exp\left(\beta \left\langle \overline{A}^{(0)}, \vartheta \right\rangle\right) + \exp\left(\beta \left\langle \overline{A}^{(1)}, \vartheta \right\rangle\right)} \tag{1}$$

where the parameter $\beta \geqslant 0$ is a measure of the *deliberateness* of the rater's decision: $\beta = 0$ means the rater's decisions are uniformly random, whereas as $\beta \to \infty$, its decisions pick the maximum of the reward under the two actions. The parameter $\vartheta \sim N\left(\theta, \mathbf{I}_d/\lambda^2\right)$ ($\mathbf{I}_d$ is a $d \times d$ identity matrix) is the rater's estimate of the true reward model and the parameter $\lambda$ is a measure of its *knowledgeability* of it, i.e., as $\lambda \to \infty$, $\vartheta \to \theta$. Alternatively, in the adaptation scenario where the online learning phase is used to align with the desires of a single user, the knowledgeability parameter can be seen as controlling the degree of alignment between the user and the general population from which preferences are aggregated. Denoting the online dataset collected by time $t$ as $\mathcal{H}_t = \{(A_t, R_t)\}_{s=1}^{t}$, we have $\mathcal{D}_t = \mathcal{D}_0 \cup \mathcal{H}_t$, the entire dataset available at time $t$.

**Notion of Regret.** Given an offline preference dataset $\mathcal{D}_0$ and an arbitrary environment $\theta$, the Bayesian Regret for $T$ rounds is given by:

$$\mathcal{BR}_T(\pi) := \sum_{t=1}^{T} \mathbb{E}_{\pi, \theta, \mathcal{D}_0}\left[\langle A^{\star}, \theta \rangle - R_t\right], \tag{2}$$

where expectation is taken over $(\pi, \theta, \mathcal{D}_0)$, and $A^{\star}(\theta) = \operatorname{argmax}_{a \in \mathcal{A}}\langle a, \theta \rangle$ (the optimal action for environment $\theta$), and $\pi$ is a policy that maps past observations $\mathcal{D}_t$ to a distribution over actions. Here, we assume that the prior distribution over the environment $\theta$ is a Gaussian distribution $\nu_0 = \mathcal{N}(\mu_0, \Sigma_0)$. To distinguish from the "informed prior" learned from $\mathcal{D}_0$, we call $\nu_0$ as the uninformed prior. The goal then is to learn a policy $\pi$ that minimizes the Bayesian regret in Equation (2).

## 3 Introducing the Preference-Warmed Posterior Sampling Algorithm

The online learning problem for Bayesian regret-minimization (in Equation (2)) that we have introduced in the previous section has two novel elements: an offline dataset to begin with, and such a dataset having only

---

**Algorithm 1** Preference-Warmed Posterior Sampling (warmPref − PS)

---

1: **Input:** Action set $\mathcal{A}$, uninformed prior $\nu_0$ over environment $\theta$, offline preference dataset $\mathcal{D}_0$.
2: Construct informed prior $\nu_1$ from $\mathcal{D}_0$ using Equation (3).
3: **for** $t = 1, 2, \ldots, T$ **do**
4:     Sample $\widehat{\theta}_t \sim \nu_t$ to take action $A_t = \mathrm{argmax}_{a \in \mathcal{A}} a^T \widehat{\theta}_t$ and receive reward $R_t$.
5:     Update dataset $\mathcal{D}_t$ and posterior $\nu_{t+1} \leftarrow \mathbb{P}(\cdot \mid \mathcal{D}_t)$ using Equation (4).
6: **end for**

---

(noisy) preference feedback generated by a human rater with limited capacity, instead of reward feedback. We adopt the posterior sampling (PS) approach to designing online bandit learning algorithms since they have a natural structure, and also because they usually offer superior performance as compared to optimism-based algorithms (Russo et al., 2018). We refer the readers to Scott (2010) and Russo et al. (2018) for an overview and background on Bayesian methods for multi-armed bandits. Thus, we introduce warmPref − PS (as Algorithm 1), a (Bayesian) posterior sampling algorithm warm-started with offline preference data. As we will see below, most of the steps are common with any meta-PS algorithm.

1. **Constructing an informed prior.** Using the offline dataset $\mathcal{D}_0$, construct an informed prior $\nu_1$,

$$\nu_1(\theta) := P(\theta \mid \mathcal{D}_0) \propto P(\mathcal{D}_0 \mid \theta) \cdot \nu_0(\theta) \propto \left[ \prod_{n=1}^{N} P(Y_n \mid \overline{A}_n^{(0)}, \overline{A}_n^{(1)}, \theta) \cdot P(\overline{A}_n^{(0)}) \cdot P(\overline{A}_n^{(1)}) \right] \cdot \nu_0(\theta) \qquad (3)$$

where $\nu_0$ is the uninformed prior and the second step follows from Equation (1) and the fact that in the context of RLHF, outputs (actions) are conditionally independent given the prompt. It is worth emphasizing here that the actions in the offline dataset carry information about the environment through the term $P(\overline{A}_n^{(\cdot)} \mid \theta)$, which incorporates information about the expert's policy, and thus improves the informativeness of the prior distribution.

2. **Online decision making.** At time $t$, get sample $\widehat{\theta}_t \sim \nu_t$, take action $A_t = \mathrm{argmax}_{a \in A} \langle a, \widehat{\theta}_t \rangle$, observe reward $R_t$, and update the dataset as $\mathcal{D}_t = \mathcal{D}_{t-1} \cup \{(A_t, R_t)\}$.

3. **Updating knowledge of the environment.** At time $t$, the environment parameter $\theta$ will have distribution $\nu_t(\theta)$, and we update our posterior as,

$$\nu_{t+1}(\theta \mid \mathcal{D}_t) \propto P(\{(A_t, R_t)\} \mid \mathcal{D}_{t-1}, \theta) \cdot \nu_t(\theta \mid \mathcal{D}_{t-1}) = P(R_t \mid A_t, \theta) \cdot P(A_t \mid \mathcal{D}_{t-1}) \cdot \nu_t(\theta \mid \mathcal{D}_{t-1}), \quad (4)$$

where $P(R_t \mid A_t, \mathcal{D}_{t-1}, \theta) = P(R_t \mid A_t, \theta)$ and $P(A_t \mid \mathcal{D}_{t-1}, \theta) = P(A_t \mid \mathcal{D}_{t-1})$. The posterior of $\vartheta$ also changes, and hence, $\vartheta_{t+1} \sim \mathcal{N}(\theta_{t+1}, \mathbf{I}/\lambda^2)$ with $\theta_{t+1} \sim \nu_{t+1}(\theta)$. We regard $\beta$ to be a known parameter. We relax this in Section 6.

*Remark* 3.1. In equation (3), we construct an informed prior using the offline preference dataset. This step can be intractable. Similarly, the posterior update of Equation (4) is also usually intractable, unless the distributions we are working with have a conjugacy property. In which case, we resort to various approximations. In Section 5, we present a practical version of this algorithm by introducing a loss function that approximates Steps 2 and 5 of Algorithm 1. This loss function is independent of the size of the action space, and hence, is extendable to infinitely-many armed bandit settings as well.

## 4 Analysis of warmPref − PS

We now present an analysis of the warmPref − PS algorithm in two steps. First, in Section 4.1, we present an "informativeness" analysis of the offline preference data $\mathcal{D}_0$, which establishes a sample complexity result for $\mathcal{D}_0$ to be informative about the optimal action. Then, based on this result, we develop an upper bound on the Bayesian regret for warmPref − PS in Section 4.2.

### 4.1 Informativeness of Offline Preference Data

We first introduce the notion of *informativeness* of the offline preference data, which characterizes how much information about the optimal action is provided by this offline preference dataset. Specifically, for purpose of analysis, we construct an 'information' set $\mathcal{U}_{\mathcal{D}_0} \subseteq \mathcal{A}$ such that it contains the optimal action with high probability (see Appendix A.1, A.2, and A.3 for details). This is useful for the analysis during the online phase; intuitively, during the online phase, $\mathsf{warmPref} - \mathsf{PS}$ is expected to sample most actions from $\mathcal{U}_{\mathcal{D}_0}$.

**Definition 4.1.** Consider a random set $\mathcal{U}_{\mathcal{D}_0} \subseteq \mathcal{A}$ measurable with respect to $\mathcal{D}_0$. For any $\epsilon \in [0, 1]$, we say $\mathcal{U}_{\mathcal{D}_0}$ is $(1 - \epsilon)$-*informative* if $P(A^\star \in \mathcal{U}_{\mathcal{D}_0}) \geqslant 1 - \epsilon$, i.e., it contains the optimal action with high probability.

This information set $\mathcal{U}_{\mathcal{D}_0}$ has to be measurable with respect to $\mathcal{D}_0$ (i.e., conditionally deterministic given $\mathcal{D}_0$). Intuitively, the offline dataset $\mathcal{D}_0$ is useful in determining the optimal action $A^\star$ if there exists a $\mathcal{U}_{\mathcal{D}_0}$ measurable to $\mathcal{D}_0$ such that (i) $\mathcal{U}_{\mathcal{D}_0}$ is $(1 - \epsilon)$-informative, and (ii) $\mathbb{E}[|\mathcal{U}_{\mathcal{D}_0}|]$ is small. In other words, one can construct a $\mathcal{U}_{\mathcal{D}_0}$ based on $\mathcal{D}_0$ such that $\mathcal{U}_{\mathcal{D}_0}$ has a small expected cardinality and contains $A^\star$ with high probability. We first present a sample complexity result (i.e., how large the offline dataset size needs to be) on $\mathcal{D}_0$ such that the set $\mathcal{U}_{\mathcal{D}_0}$ constructed in the appendix is $(1 - \epsilon)$-informative. We start by studying the special case of the set $\mathcal{U}_{\mathcal{D}_0}$ being a singleton to elucidate its dependence on various parameters, but discussion in the next section will not require this assumption. The result below shows this dependence, i.e., how large does $\mathcal{D}_0$ need to be such that $\mathsf{warmPref} - \mathsf{PS}$ can infer $A^\star$ from it.

**Theorem 4.2.** *Let the action set $\mathcal{A}$ have size $K$ with a sampling distribution $\mu$ such that $0 < \mu_{min} \leqslant \mu_k \leqslant \mu_{max} < 1$, $\forall k \in [K]$. Given some $\epsilon \in (0, 1)$ and finite $\beta < \infty$, let $\lambda \to \infty$. Then, the singleton set $\mathcal{U}_{\mathcal{D}_0} = \{A^\star\}$ is $(1 - \epsilon)$-informative if*

$$N > N_0 := \frac{\ln K + (k_{max} - 1)\ln \ln K}{\mu_{min}^2 \epsilon}, \quad where \tag{5}$$

$$k_{max} = \max_{i,j \in [K]} \frac{\ln\left(\left(\frac{2K^2}{\epsilon} - 1\right)\left(\frac{1}{\Phi(x_{i,j})} - 1\right)\right)}{\beta\langle a_i - a_j, \theta_0\rangle}, \, and \, x_{i,j} = \frac{(a_i - a_j)^T \mu_0}{\sqrt{(a_i - a_j)^T \Sigma_0 (a_i - a_j)}},$$

*and, $N$ is the size of the preference dataset and $\Phi(\cdot)$ is the standard Normal CDF.*

The proof can be found in Appendix A.3. The above theorem provides a bound on the size of the offline preference dataset $\mathcal{D}_0$ needed to find a singleton information set $\mathcal{U}_{\mathcal{D}_0}$ containing the optimal action. To understand the above result, note that in the special case of $K = 2$, we have the following.

**Corollary 4.3.** *For an action set $\mathcal{A} = \{a_0, a_1\}$, any $\beta \in (0, \infty)$ and $\epsilon \in (0, 1)$, with $\lambda \to \infty$, if*

$$N > N_0 := \frac{\ln\left(\left(\frac{1}{\epsilon} - 1\right)\left(\frac{1}{\Phi(x)} - 1\right)\right)}{\beta\langle a_0 - a_1, \theta_0\rangle}, \tag{6}$$

*where $x := \frac{(a_0 - a_1)^T \mu_0}{\sqrt{(a_0 - a_1)^T \Sigma_0 (a_0 - a_1)}}$, then there exists a singleton $\mathcal{U}_{\mathcal{D}_0}$ that is $(1 - \epsilon)$-informative.*

See Appendix A.1 and A.1.4 for proof. The above corollary reveals how the offline dataset size $N$ needed to infer the optimal action depends on the deliberateness parameter in the presence of noisy comparisons. We see that as $\beta \to \infty$, we have $N_0 \to 0$, i.e., we only need a single comparison.

### 4.2 Regret Bound

We now first introduce an information theoretic result from the literature for Bayesian regret for a posterior sampling algorithm. The discussion below is for the general case where the information set $\mathcal{U}_{\mathcal{D}_0}$ is not necessarily a singleton.

**Theorem 4.4.** *(Hao et al. (2023)) For any $(1 - \epsilon)$-informative set $\mathcal{U}_{\mathcal{D}_0} \subseteq \mathcal{A}$, the Bayesian Regret of any $\mathsf{PS}$ algorithm can be upper bounded as:*

$$\mathcal{BR}_T(\mathsf{PS}) \leqslant \sqrt{T\mathbb{E}[|\mathcal{U}_{\mathcal{D}_0}|]\ln(\mathbb{E}[|\mathcal{U}_{\mathcal{D}_0}|])} + \epsilon\ln(K/\epsilon) + C_1 T\epsilon,$$

*where $C_1$ is the bound on the expected reward range, i.e., $\mathbb{E}[\max a^T \theta] - \mathbb{E}[\min a^T \theta] \leqslant C_1$.*

To apply the above theorem, we can construct the set $\mathcal{U}_{\mathcal{D}_0}$ in the following way: it contains all actions that have been preferred to another action *at least* once in the offline dataset $\mathcal{D}_0$ and also includes any actions that do not appear in the dataset $\mathcal{D}_0$ [1]. Thus, $\mathcal{U}_{\mathcal{D}_0}$ can contain upto $K$ actions. For clarity, $\mathcal{U}_{\mathcal{D}_0}$ is constructed as the set of all actions that are not ruled out by the offline comparisons. An action is included if it (i) wins at least one comparison in $\mathcal{D}_0$, or (ii) never appears in $\mathcal{D}_0$ at all, in which case the data provide no evidence against it. This makes $\mathcal{U}_{\mathcal{D}_0}$ a conservative set of actions that remain compatible with being optimal after observing $\mathcal{D}_0$.

Now, let $\Delta := \ln(T\beta)/\beta$, $\alpha_1^\Delta := K \min(1, \Delta)$, and $\alpha_2 := \lambda^{-1}\sqrt{2\ln(2d^{1/2}T)}$. Then, denote

$$
\begin{aligned}
\widetilde{f}_1 &:= \left(1 - \frac{1}{1 + \exp\left(\beta\left(\min(1, \Delta) + \alpha_2 - \alpha_1^\Delta\right)\right)}\right)^N + (1 - \mu_{\min})^{2N} \quad, \ f_1 = \widetilde{f}_1 + \frac{1}{T} \ , \\
f_2 &:= \min\left(\left(\alpha_1^\Delta\right)^2 + \frac{NK}{T\beta}\left(1 + \exp\left(-\beta\alpha_2 + \alpha_1^\Delta\right)\right)^{-N} + \frac{2}{T} \ , \ K\right) \ .
\end{aligned}
\tag{7}
$$

The constants $(f_1, f_2)$ characterize the offline preference dataset in terms of its size $N$, and expert's competence parameters $\lambda$ and $\beta$. Then, we have the following guarantee on the informativeness and size of the set $\mathcal{U}_{\mathcal{D}_0}$.

**Lemma 4.5.** *If $\mu_{min} > 0$, then set $\mathcal{U}_{\mathcal{D}_0}$ constructed above is $(1 - f_1)$-informative, and $\mathbb{E}[|\mathcal{U}_{\mathcal{D}_0}|] \leqslant f_2$.*

*Proof sketch.* We upper bound the probability of the optimal action $A^\star$ not being in $\mathcal{U}_{\mathcal{D}_0}$ (the 'error probability'), and by defining an event $\mathcal{E}_{(n)} := \{\langle A^\star - a_n, \theta\rangle \leqslant \Delta\}$, where $(A^\star, a_n)$ is the action tuple in $\mathcal{U}_{\mathcal{D}_0}$ for some $n \in [N]$ and $\Delta \in \mathbb{R}$. Then, we decompose this error probability based on the event $\mathcal{E}_{(n)}$ and the sub-optimality gap of actions ($\Delta$). To bound the expected cardinality of $\mathcal{U}_{\mathcal{D}_0}$, we again decompose based on $\mathcal{E}_{(n)}$ and use a Poisson approximation to bound the probability. See Appendix A.4 for the complete proof. $\square$

Here, $\alpha_1^\Delta$ and $\alpha_2$ are representative of the nature of the problem, and take into the consideration the information loss due to finite deliberateness and knowledgeability of the rater respectively. Both $\alpha_1^\Delta, \alpha_2 \to 0$, as $\beta$ and $\lambda$ get large. The parameter $f_1$ captures the error probability of the optimal action not being in $\mathcal{U}_{\mathcal{D}_0}$, and decays exponentially as the size of the dataset increases. Note that we need $\mu_{\min}$ and $\mu_{\max} \in (0, 1)$ to obtain a nonzero sampling probability over all the actions.

Now using Lemma 4.5 in conjunction with the information theoretic upper bound in Theorem 4.4, we obtain the following main result about the Bayesian regret of the warmPref $-$ PS algorithm:

**Theorem 4.6.** *The Bayesian regret of the warmPref $-$ PS algorithm can be bounded as*

$$
\mathcal{BR}_T(\pi_{\mathsf{warmPref-PS}}) \leqslant \underbrace{\sqrt{T f_2 \left(\ln(f_2) + f_1 \ln\left(K/f_1\right)\right)}}_{main\ term} + 2\sqrt{2\ln(K)} T \left(\widetilde{f}_1 + \frac{1}{T}\right)
$$

The proof can be found in Appendix A.4. Although the bound in Theorem 4.6 appears to be linear in $T$, in fact $\widetilde{f}_1 < 1$, and as the dataset size $N \to \infty$, $\widetilde{f}_1 \to 0$, implying that the second term behaves like a constant. Second, for the main term, as the deliberateness $\beta$ and dataset size $N$ increase, the information ratio ($f_2$) decreases exponentially and then the entropy part ($\ln(f_2) + f_1 \ln(K/f_1)$) decreases further until $f_2 = 1$. Finally, note that as the preference dataset parameters $\lambda$ and $\beta$ get large, the main term in the regret bound above converges to 0. Thus, the algorithm has constant regret in the case where the offline dataset is very large and is from a near-optimal expert. Note here that the algorithm currently requires the knowledge of rater competence ($\lambda$ and $\beta$) to achieve the regret bound above. Nevertheless, in the following sections we will discuss methods to estimate these parameters from the offline dataset.

*Remark* 4.7. Note that $f_2 \leqslant K$, so $\mathcal{U}_{\mathcal{D}_0}$ cannot grow arbitrarily large. Second, the informativeness of the offline dataset depends on both the dataset size $N$ and its quality (measured by $\beta$). When both $N$ and $\beta$ go to infinity, our regret bound in Theorem 4.6 reduces to $\mathcal{O}(\sqrt{\ln(K) + \ln(T)})$, which is sublinear. Thus, our

---

[1]Note that this construction is an algorithmic choice that integrates well with the posterior sampling style of algorithms. There can be other construction criteria of $\mathcal{U}_{\mathcal{D}_0}$ as well.

---

**Algorithm 2** Bootstrapped warmPref − PS

---
1: **Input:** Horizon $T$, offline preference dataset $\mathcal{D}_0$, action set $\mathcal{A}$, knowledgeability $\lambda$, deliberateness $\beta$.
2: **for** $t = 1, 2, \ldots, T$ **do**
3:    Sample a set of perturbations $\mathcal{P}_t = \{(\zeta_s, \omega_n, \theta', \vartheta')\}$.
4:    Solve Equation (9) using this set $\mathcal{P}_t$ to find $(\widehat{\theta}_t, \widehat{\vartheta}_t)$.
5:    Take action $A_t = \text{argmax}_{a \in \mathcal{A}} \langle a, \widehat{\theta}_t \rangle$, receive reward $R_t$, and update $\mathcal{D}_t \leftarrow \mathcal{D}_{t-1} \cup \{(A_t, R_t)\}$.
6: **end for**

---

result shows that when the offline dataset has both high quality and large size, the per unit regret of the proposed algorithm is negligibly small. Finally, note that in the asymptotic sense this upper bound matches the lower bound in the classical linear bandit setting with no offline data and bandit feedback.

*Remark* 4.8. Theorem 4.6 depends on how informative the offline preferences are, which is controlled by the rater competence $(\lambda, \beta)$. Lemma 4.5 shows that the probability of excluding the optimal action decreases exponentially in $N$, with faster decay when $\beta$ and $\lambda$ are larger. Thus sublinear regret does not require $N \to \infty$ in practice. Even moderate $N$ and moderately competent raters produce a small, high-probability set $\mathcal{U}_{\mathcal{D}_0}$, which, as we will see in Section 6, is consistent with our empirical results.

*Remark* 4.9. The current analysis has been done in the finite armed bandit setting for tractability reasons and practical relevance to RLHF with finite vocabulary and context-output pair sizes, and we leave the analysis of infinite-many armed bandit setting for future work. Nevertheless, in the next section we present an approximate loss function for the warmPref − PS algorithm that works for the infinite armed setting as well and we will see that it performs substantially better than available baselines (in the finite-arm setting).

## 5  A Practical Approximation of the warmPref − PS Algorithm

As mentioned before, the posterior update in Equation (3) and Equation (4) lacks the conjugacy property due to the $P(A_t \mid \mathcal{D}_{t-1})$ term, and is hence intractable. However, it inspires us to design a practical algorithm in the manner of well established Bayesian bootstrapping ideas (Osband et al., 2019) where a surrogate loss function is constructed with added noise, and is then optimized to obtain the Maximum A Posteriori (MAP) estimate. This provides a point estimate of the unknown parameters $(\theta, \vartheta)$, but due to the added noise can be viewed as a sample from an approximation to the posterior distribution.

**A surrogate loss function.** We start with the MAP estimate problem for $(\theta, \vartheta)$ given the offline and online dataset $\mathcal{D}_{t-1}$ at time $t - 1$. We show that this is equivalent to minimizing a particular surrogate loss function as described in the lemma below:

**Lemma 5.1.** *At time $t$, the MAP estimate of $(\theta, \vartheta)$ can be constructed by solving the following equivalent optimization problem:*

$$(\theta_{opt}, \vartheta_{opt}) = \underset{\theta, \vartheta}{\text{argmax}} \, P(\theta, \vartheta \mid \mathcal{D}_{t-1}) \equiv \underset{\theta, \vartheta}{\text{argmin}} \, \mathcal{L}_1(\theta, \vartheta) + \mathcal{L}_2(\theta, \vartheta) + \mathcal{L}_3(\theta, \vartheta),$$

$$where, \qquad \mathcal{L}_1(\theta, \vartheta) := \frac{1}{2} \sum_{s=1}^{t-1} \left(R_s - \langle A_s, \theta \rangle\right)^2,$$

$$\mathcal{L}_2(\theta, \vartheta) := -\sum_{n=1}^{N} \beta \langle \overline{A}_n^{(Y_n)}, \vartheta \rangle + \ln\left(e^{\beta \langle \overline{A}_n^{(0)}, \vartheta \rangle} + e^{\beta \langle \overline{A}_n^{(1)}, \vartheta \rangle}\right),$$

$$\mathcal{L}_3(\theta, \vartheta) := \frac{\lambda^2}{2} ||\theta - \vartheta||_2^2 + \frac{1}{2}(\theta - \mu_0)^T \Sigma_0^{-1}(\theta - \mu_0).$$

(8)

See Appendix A.5 for proof. A close look at Equation (8) shows that $\mathcal{L}_1$ captures the likelihood of the online rewards, $\mathcal{L}_2$ captures the likelihood of preferences from the offline preference dataset $\mathcal{D}_0$, and $\mathcal{L}_3$ handles the prior distribution of $\theta$ and $\vartheta$. We could also regard $\beta$ to be unknown but that leads to a non-convex loss function. So, we estimate that separately in Section 6 and then plug it in Equation (8). Minimizing the above loss function however, only yields a point estimate of $(\theta, \vartheta)$ that is deterministic given the dataset $\mathcal{D}_{t-1}$.

**Perturbing the loss function.** As mentioned above, the idea now is to *perturb* the loss function in Equation (8) with some noise, so that the MAP point estimates we get from this perturbed surrogate loss function serve as *samples* from a distribution that approximates the true posterior (Osband et al., 2019; Lu & Van Roy, 2017; Qin et al., 2022; Dwaracherla et al., 2022). To that end, we propose a perturbation of the 'online' loss function $\mathcal{L}_1(\cdot)$ by additive Gaussian noise, of the 'offline' loss function $\mathcal{L}_2(\cdot)$ by multiplicative random weights, and of the 'prior' loss function $\mathcal{L}_3(\cdot)$ by random samples from the prior distribution as follows: (i) *Online perturbation.* Let $\zeta_s \sim \mathcal{N}(0,1)$, all i.i.d. Then, the perturbed $\mathcal{L}_1(\cdot)$ becomes $\mathcal{L}_1'(\theta,\vartheta) = \frac{1}{2}\sum_{s=1}^{t-1}\left(R_s + \zeta_s - \langle A_s, \theta\rangle\right)^2$, (ii) *Offline perturbation.* Let $\omega_n \sim \text{Bern}(0.5)$, all i.i.d. Then, the perturbed $\mathcal{L}_2(\cdot)$ becomes $\mathcal{L}_2'(\theta,\vartheta) = -\sum_{n=1}^{N}\omega_n\left[\beta\langle\overline{A}_n^{(Y_n)},\vartheta\rangle + \ln\left(e^{\beta\langle\overline{A}_n^{(0)},\vartheta\rangle} + e^{\beta\langle\overline{A}_n^{(1)},\vartheta\rangle}\right)\right]$, and (iii) *Prior perturbation.* Let $\theta' \sim \mathcal{N}(\mu_0, \Sigma_0)$, and $\vartheta' \sim \mathcal{N}(\mu_0, \mathbf{I}_d/\lambda^2)$, all i.i.d. Then, the perturbed $\mathcal{L}_3(\cdot)$ becomes $\mathcal{L}_3'(\theta,\vartheta) = \frac{\lambda^2}{2}||\theta - \vartheta + \vartheta'||_2^2 + \frac{1}{2}(\theta - \mu_0 - \theta')^T\Sigma_0^{-1}(\theta - \mu_0 - \theta')$. Then, at time $t$, we get the following MAP point estimate from the perturbed surrogate loss function,

$$(\widehat{\theta}_t, \widehat{\vartheta}_t) = \underset{\theta,\vartheta}{\operatorname{argmin}}\ \mathcal{L}'(\theta,\vartheta) = \underset{\theta,\vartheta}{\operatorname{argmin}}\ \mathcal{L}_1'(\theta,\vartheta) + \mathcal{L}_2'(\theta,\vartheta) + \mathcal{L}_3'(\theta,\vartheta), \tag{9}$$

which are well understood to have a distribution that approximates the actual posterior distribution. Note that the perturbed surrogate loss function is convex, can be optimized easily and is independent of the number of arms, and hence is scalable to infinitely-many armed bandit setting as well. In addition, it can be extended easily to the setting where the offline dataset comes from *multiple* experts with different $(\lambda_i, \beta_i)$ competence tuples. Specifically, for $M$ experts, there will be $M$ similar terms for $\mathcal{L}_2'(\cdot)$ and $\mathcal{L}_3'(\cdot)$ respectively, while $\mathcal{L}_1'(\cdot)$ will remain unchanged. This yields the Bootstrapped $\mathsf{warmPref-PS}$ as Algorithm 2.

*Remark* 5.2. In the next section, we will show that while a theoretical analysis of the practical approximation proposed above is challenging, it does have excellent empirical performance and can be scaled up to large problems as well.

## 6 Empirical Results

We now present results on the empirical performance of the Bootstrapped $\mathsf{warmPref-PS}$ algorithm introduced in the previous section. We are particularly interested in the following questions: (i) How much is the reduction in cumulative Bayesian regret due to warm start with an offline preference dataset? (ii) How much does the competence (in terms of $\lambda$ and $\beta$) of the expert (rater) who generated the offline preference affect regret? (iii) Is $\mathsf{warmPref-PS}$ robust to mis-specification of $\lambda$ and $\beta$?

**Baselines.** To evaluate the Bootstrapped $\mathsf{warmPref-PS}$ algorithm, we consider the following baselines: (i) (`vanilla`) `PS`, a PS algorithm that does not use the offline dataset, (ii) `LinTS` from Li et al. (2010) and Agrawal & Goyal (2013), (iii) Direct Preference Optimization (`DPO`) (Rafailov et al., 2024), and (iv) Preferential Bayesian Optimization (`PBO`) (González et al., 2017). All plots show empirical regret. Another possible baseline can be based on optimism methods to directly learn a $\widehat{\theta}$ from $\mathcal{D}_0$, and use that to warm-start the online learning. However, such optimism-based algorithms are computationally intractable as they need to construct confidence sets, and then optimize over them.

*Remark* 6.1. `DPO` and `PBO` cannot be trivially extended to our problem setting, i.e., fixed offline preference dataset with online numerical reward learning. Comparing `DPO` and `PBO` trained only on $\mathcal{D}_0$ is not fair, and hence, we consider an offline-online variant of `DPO`, called `Hybrid-DPO`, with $\epsilon$-greedy online exploration, and a learned reward model approach for `PBO`. Please see Appendix A.7 and Appendix A.8 for more details.

*Remark* 6.2. To the best of our knowledge, no other works that formalize learning from offline preference data and online numerical rewards exist, hence, there are no other baselines available in the literature. 'Hybrid bandit' studies cited in Related Work assume numeric rewards in both phases, making direct and fair comparison with $\mathsf{warmPref-PS}$ not possible. A conceivable proxy is to fit a reward model on the offline preference data, convert preferences into pseudo-rewards, and benchmark algorithms on this reward-based offline dataset. However, converting feedback modalities compromises the validity of any fair comparison.

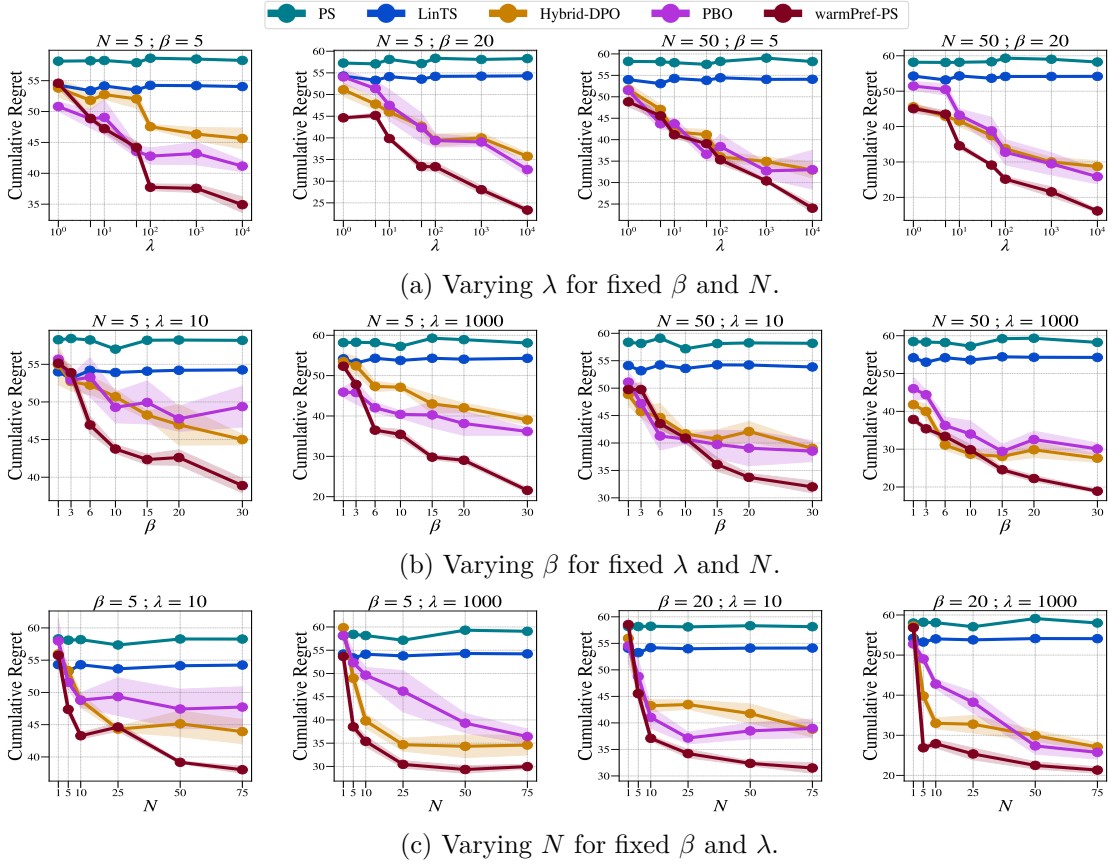

(a) Varying $\lambda$ for fixed $\beta$ and $N$.

(b) Varying $\beta$ for fixed $\lambda$ and $N$.

(c) Varying $N$ for fixed $\beta$ and $\lambda$.

Figure 2: Cumulative Regret comparison with varying $N$, $\beta$, and $\lambda$. Shaded region around the mean line represents 1 standard deviation over 5 independent runs.

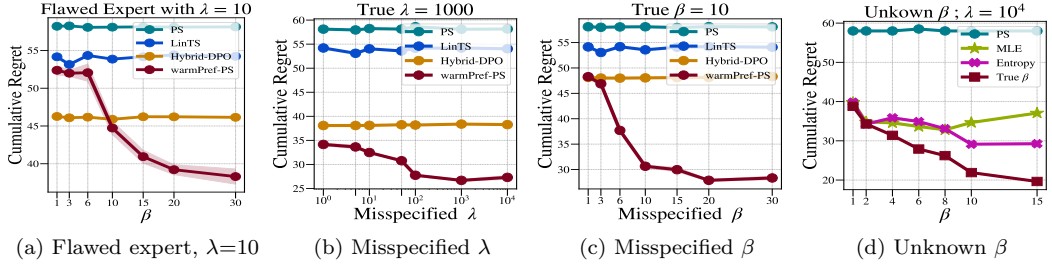

(a) Flawed expert, $\lambda$=10  (b) Misspecified $\lambda$  (c) Misspecified $\beta$  (d) Unknown $\beta$

Figure 3: Sensitivity analysis with flawed expert policy, misspecified and unknown competence.

**Evaluation protocol.** Unless specified otherwise, for all experiments, we have $K = 50$ arms, dimension $d = 6$, $\lambda = 100$, $\beta = 10$, dataset size $N = 20$, and horizon $T = 300$. We averaged over 5 runs (with random seeds). For easy interpretation, we let $\mu \sim \text{Unif}(\cdot)$.

**Value of Offline Preferences.** We first aim to understand the impact of the offline preference dataset $\mathcal{D}_0$ on the performance of warmPref $-$ PS as three parameters, $\beta$, $\lambda$ and $N$ vary. Figures 2(a) shows that as $\lambda$ increases (the expert has a better estimate of the reward model $\theta$), the regret reduces and this reduction is substantial for the warmPref $-$ PS algorithm than for the naive-PS algorithm (vanilla-PS and LinTS are unaffected by $\mathcal{D}_0$ as they do not use the offline dataset). Figure 2(b) shows that (for fixed $\lambda$ and $N$) as $\beta$ increases, the regret reduces substantially. Figure 2(c) now fixes $\beta$ and $\lambda$, and shows that as dataset size $N$ increases, even with a 'mediocre' expert ($\beta = 5$) the regret reduces substantially, and in fact by 25 to 50% even with a very small ($N = 5$) dataset size. The conclusion from these results is that even by using a

small amount of offline data from a mediocre expert, the Bootstrapped $\mathsf{warmPref} - \mathsf{PS}$ algorithm achieves significant reduction in regret over the baselines.

**Sensitivity to parameter specification errors.** The (Bootstrapped) $\mathsf{warmPref} - \mathsf{PS}$ algorithm in Section 5 requires a knowledge of expert's parameters $\beta$ and $\lambda$. In Figure 3, we study the sensitivity of the algorithm's performance to errors in specification of these parameters (as well as of assuming a Bradley-Terry model for the rater). Due to space constraints, further ablation studies are provided in Appendix A.9.

(i) **Different Preference Generation Expert Policy.** Though the learning agent assumes Equation (1) as the expert's generative model, we consider it to actually use a deterministic greedy policy. Actions $\overline{A}_n^{(0)}$ and $\overline{A}_n^{(1)}$ are sampled, and then choose $Y_n = \mathrm{argmax}_{i \in \{0,1\}} \beta \langle \overline{A}_n^{(i)}, \vartheta \rangle$, where $\vartheta \sim \mathcal{N}(\theta, \mathbf{I}_d/\lambda^2)$. In Figure 3(a), we see that even when the learning agent's assumption of the expert policy is *flawed*, $\mathsf{warmPref} - \mathsf{PS}$ outperforms the baselines.

(ii) **Misspecified Competence parameters.** First, we generate the offline data with the true $\lambda = 10^3$ but the algorithm uses a misspecified $\lambda$. Second, we generate the offline data with the true $\beta = 10$ but the algorithm uses a misspecified $\beta$. Figure 3(b) and 3(c) show that although the performance of $\mathsf{warmPref} - \mathsf{PS}$ decreases as the degree of flawness increases, our algorithm still outperforms the baselines.

(iii) **Unknown Competence.** As seen in Section 5, Bayesian bootstrapping requires an input for the competence level. In practice, this is not available but, can be estimated from the offline dataset. There are many ways of estimating $\beta$ (Beliaev et al., 2022a; Hao et al., 2023), but the most common methods are: (i) Maximum Likelihood Estimation (MLE) : optimize $\beta$ over the negative log-likelihood of the $\mathcal{D}_0$ and, (ii) Entropy : calculate entropy of the empirical distribution of the actions occuring in $\mathcal{D}_0$, call it $\mathcal{H}_{\mathcal{D}_0}$ and use $\hat{\beta} = c/\mathcal{H}_{\mathcal{D}_0}$, where $c > 0$ is a hyperparameter. We compare entropy-based method and MLE based method for $\mathsf{warmPref} - \mathsf{PS}$ with baselines: (1) use true $\beta$ with $\mathsf{warmPref} - \mathsf{PS}$ and, (2) vanilla PS. To isolate the effect of $\beta$, we let $\lambda = 10^4$. As shown in Figure 3(d), although performance degrades due to estimation, $\mathsf{warmPref} - \mathsf{PS}$ still beats baselines.

# 7 Conclusion

In this paper, we proposed $\mathsf{warmPref} - \mathsf{PS}$, a Bayesian posterior sampling-based algorithm that efficiently incorporates offline preference data to warm-start the online learning phase. We provide theoretical foundations for bridging the gap between fixed, offline preferences and online learning. We further introduced Bootstrapped $\mathsf{warmPref} - \mathsf{PS}$, a computationally tractable extension designed to handle large-scale environments. Our theoretical results and empirical evaluations demonstrate the robustness and superior performance of $\mathsf{warmPref} - \mathsf{PS}$. While additional work is needed to refine our approach for RLHF, we provide a promising foundation for further development in this space.

# Impact Statement

Although our study is theoretical and primarily uses synthetic data, the setting is relevant to RLHF and human-in-the-loop systems. Using offline preference datasets in such applications raises concerns when raters are imperfect or unrepresentative, potentially introducing or amplifying bias. Real-world deployments would also need to ensure proper data provenance, rater consent, and privacy. Further, competence modeling could unintentionally weight certain human feedback more than others, affecting fairness and accountability. Hence, future applied systems should use diverse raters, fairness-aware weighting, and transparent preference-collection practices to mitigate these risks.

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

# A    Appendix

This appendix is structured as follows.

| | |
|---|---|
| Section A.1 | First building block of main result Theorem 4.2. Deals with two actions and understanding comparison noises. Contains Lemma A.1 and proofs. |
| Section A.2 | Second building block. Deals with multiple actions but no comparison noises i.e. $\beta \to \infty$. Contains Lemma A.2 and proofs. |
| Section A.3 | Final building block. Combines the results from Lemmas A.1 and A.2, and gives proof of Theorem 4.2. |
| Section A.4 | Concerns regret analysis, and contains proofs of Lemma 4.5 given in Lemmas A.3 and A.4. |
| Section A.5 | Contains details on Bayesian bootstrapping of warmPref − PS and proof of Lemma 5.1. |
| Section A.6 | Gives proof of concept of `warmTSOF` algorithm and experimental results. |
| Section A.7 | Gives training details of DPO and IPO. |
| Section A.9 | More ablation studies continued from Section 6. |

### A.1 Understanding Two Actions and Finite Deliberateness

In the building block towards Theorem 4.2, we consider the case with $K = 2$ and hence $\mathcal{A} = \{a_0, a_1\}$, which means $\overline{A}_i^{(0)} = a_0$ and $\overline{A}_i^{(1)} = a_1$ for all $i \in [N]$. Here, we focus on understanding how the comparison noises, due to finite deliberateness, affect the results. In other words, we see how the effect of deliberateness results in more than one sample being required to determine the optimal action with high probability.

Essentially, given an offline dataset $\mathcal{D}_0$, we construct a *warm*-posterior of the likelihood that an action is optimal. Based on this posterior over the actions, we can construct an action subset $\mathcal{U}_{\mathcal{D}_0} \subset \mathcal{A}$ with $|\mathcal{U}_{\mathcal{D}_0}| = 1$.

#### A.1.1 Constructing the Algorithm

In this part, we deal with the question that given an offline dataset $\mathcal{D}_0$, how to develop an algorithm to constructing an action subset $\mathcal{U} \subset \mathcal{A}$ with $|\mathcal{U}| = 1$.

For this, we need to calculate the posterior distribution of an action being optimal given the offline dataset $\mathcal{D}_0$. If $P(a_0 = A^\star \,|\, \mathcal{D}_0) > P(a_1 = A^\star \,|\, \mathcal{D}_0)$, then $\mathcal{U} = \{a_0\}$, else $\mathcal{U} = \{a_1\}$. Let $p_0 := \exp\left(\beta \langle a_0, \vartheta \rangle\right)$ and $p_1 := \exp\left(\beta \langle a_1, \vartheta \rangle\right)$. So,

$$
\begin{aligned}
P(a_0 = A^\star \,|\, \mathcal{D}_0) &= \frac{P(\mathcal{D}_0 \,|\, a_0 = A^\star) \cdot P(a_0 = A^\star)}{P(\mathcal{D}_0)} \\[2mm]
&= \frac{\int P(\mathcal{D}_0 \,|\, a_0 = A^\star \,;\, \vartheta)\, d\vartheta \cdot P(a_0 = A^\star)}{\int\int P(\mathcal{D}_0 \,|\, \beta, \lambda)\, d\lambda d\beta} \\[2mm]
&= \frac{\int \left(\dfrac{p_0}{p_0 + p_1}\right)^N d\vartheta \cdot P(a_0 = A^\star \,;\, \theta_0)}{\int\int P(\mathcal{D}_0 \,|\, \beta, \lambda)\, d\lambda d\beta} \qquad\qquad \left(\theta_0 \sim \mathcal{N}(\mu_0, \Sigma_0)\right) \\[2mm]
&= \frac{\int \left(\dfrac{p_0}{p_0 + p_1}\right)^N d\vartheta \cdot P\left(\langle a_0, \theta_0 \rangle \geqslant \langle a_1, \theta_0 \rangle\right)}{\int\int P(\mathcal{D}_0 \,|\, \beta, \lambda)\, d\lambda d\beta} \\[2mm]
&= \frac{\int \left(\dfrac{p_0}{p_0 + p_1}\right)^N d\vartheta}{\int\int P(\mathcal{D}_0 \,|\, \beta, \lambda)\, d\lambda d\beta} \cdot P\left(\langle a_0 - a_1, \theta_0 \rangle \geqslant 0\right) \\[2mm]
&= \frac{\int \left(\dfrac{p_0}{p_0 + p_1}\right)^N d\vartheta}{\int\int P(\mathcal{D}_0 \,|\, \beta, \lambda)\, d\lambda d\beta} \cdot \left(1 - \Phi\left(-\frac{(a_0 - a_1)^T \mu_0}{\sqrt{(a_0 - a_1)^T \Sigma_0 (a_0 - a_1)}}\right)\right)
\end{aligned}
$$

(10)

, where $\Phi$ is the CDF of the standard normal distribution. Similar expression follows for $P(a_1 = A^\star \,|\, \mathcal{D}_0) = 1 - P(a_0 = A^\star \,|\, \mathcal{D}_0)$.

#### A.1.2 Limiting Behaviour of the Algorithm for Optimal Expert

Here, we see that under our specified algorithm, for any offline data size $N \geqslant 1$, as $\beta, \lambda \to \infty$, $\mathcal{U} \to \{A^*\}$ almost surely. It is easy to see this. As $\lambda \to \infty$, we have $\vartheta \to \theta_0$. Then,

$$
\lim_{\beta \to \infty, \vartheta \to \theta_0} \left(\frac{p_0}{p_0 + p_1}\right)^N = \lim_{\beta \to \infty, \vartheta \to \theta_0} \left(\frac{1}{1 + e^{-\beta \langle a_0 - a_1, \vartheta \rangle}}\right)^N.
$$

Now observe that if $\beta, \lambda \to \infty$, if $a_1 = A^\star$, then $\lim_{\lambda \to \infty} \langle a_0 - a_1, \vartheta \rangle \leqslant 0 \implies \lim_{\beta, \lambda \to \infty} \left( \frac{p_0}{p_0 + p_1} \right)^N \to 0 \implies P(a_0 = A^\star \mid \mathcal{D}_0) \to 0$. Same holds for when $\beta, \lambda \to \infty$ and if $a_0$ is optimal. Which means the specified decision rule above converges with $\mathcal{U} \to \{A^\star\}$ almost surely.

### A.1.3 Limiting Behaviour of the Algorithm for Large Datasets

Here, we show that under our specified algorithm, for any finite $\beta > 0$, as $\lambda, N \to \infty$, $\mathcal{U} \to \{A^*\}$ almost surely.

For this, we just calculate the ratio $\lim_{\lambda, N \to \infty} \frac{P(a_0 = A^\star \mid \mathcal{D}_0)}{P(a_1 = A^\star \mid \mathcal{D}_0)}$ and check whether it tends to zero or infinity. Let $x := \frac{(a_0 - a_1)^T \mu_0}{\sqrt{(a_0 - a_1)^T \Sigma_0 (a_0 - a_1)}}$. So then,

$$
\begin{aligned}
\lim_{\lambda, N \to \infty} \frac{P(a_0 = A^\star \mid \mathcal{D}_0)}{P(a_1 = A^\star \mid \mathcal{D}_0)} &= \lim_{\vartheta \to \theta_0, N \to \infty} \frac{\left( \frac{p_0}{p_0 + p_1} \right)^N \cdot (1 - \Phi(-x))}{\left( \frac{p_1}{p_0 + p_1} \right)^N \cdot (1 - \Phi(x))} \\
&= \lim_{\vartheta \to \theta_0, N \to \infty} \left( \frac{p_0}{p_1} \right)^N \cdot \frac{\Phi(x)}{1 - \Phi(x)} \\
&= \lim_{N \to \infty} [\exp(\beta \langle a_0 - a_1, \theta_0 \rangle)]^N \cdot \frac{\Phi(x)}{1 - \Phi(x)}
\end{aligned}
$$

Now, we can apply the same argument of, if $a_0 = A^\star$, then $\langle a_0 - a_1, \theta_0 \rangle \geqslant 0$ to see that the above expression tends to positive infinity for any finite $\beta > 0$. Hence, we can construct $\mathcal{U} = \{A^\star\}$ almost surely.

### A.1.4 Sample Complexity for Finite Deliberateness

In this part, we consider for any finite $\beta > 0$, as $\lambda \to \infty$, and any given $\epsilon \in (0, 1)$, under our specified algorithm, how large does $N$ need to be to ensure $P(\mathcal{U} = \{A^*\}) \geqslant 1 - \epsilon$?

**Lemma A.1.** *For an action set $\mathcal{A} = \{a_0, a_1\}$ and any finite $\beta \in (0, \infty)$, with $\lambda \to \infty$ and for some $\epsilon \in (0, 1)$, the size of the offline dataset to ensure $\mathcal{U}_{\mathcal{D}_0} = \{A^\star\}$ and hence $(1 - \epsilon)$-informative is:*

$$
N \geqslant \frac{\ln \left( \left( \frac{1}{\epsilon} - 1 \right) \left( \frac{1}{\Phi(x)} - 1 \right) \right)}{\beta \langle a_0 - a_1, \theta_0 \rangle}, \tag{11}
$$

*where $x := \frac{(a_0 - a_1)^T \mu_0}{\sqrt{(a_0 - a_1)^T \Sigma_0 (a_0 - a_1)}}$, and $\Phi(\cdot)$ is the standard Normal CDF.*

*Proof.* Assume $A^\star = a_0$. Then, we want $P(a_0 = A^\star \mid \mathcal{D}_0) > 1 - \epsilon$ and $P(a_1 = A^\star \mid \mathcal{D}_0) < \epsilon$. Letting Let $x := \frac{(a_0 - a_1)^T \mu_0}{\sqrt{(a_0 - a_1)^T \Sigma_0 (a_0 - a_1)}}$ same as before and taking the ratio of these as $\lambda \to \infty$ for a finite $\beta, N > 0$, we have

$$
\begin{aligned}
\lim_{\lambda \to \infty} \frac{P(a_0 = A^\star \mid \mathcal{D}_0)}{P(a_1 = A^\star \mid \mathcal{D}_0)} &= \lim_{\vartheta \to \theta_0} \frac{\left( \frac{p_0}{p_0 + p_1} \right)^N \cdot (1 - \Phi(-x))}{\left( \frac{p_1}{p_0 + p_1} \right)^N \cdot (1 - \Phi(x))} && > \frac{1 - \epsilon}{\epsilon} \\
&= [\exp(\beta \langle a_0 - a_1, \theta_0 \rangle)]^N \cdot \frac{\Phi(x)}{1 - \Phi(x)} && > \frac{1}{\epsilon} - 1 \\
&\implies N > \frac{\ln \left( \left( \frac{1}{\epsilon} - 1 \right) \left( \frac{1}{\Phi(x)} - 1 \right) \right)}{\beta \langle a_0 - a_1, \theta_0 \rangle}.
\end{aligned} \tag{12}
$$

Without loss of generality, similar argument holds for if $A^\star = a_1$. $\qquad \square$

## A.2 Understanding Multiple Actions and Infinite Deliberateness

In this building block towards Theorem 4.2, we focus on the case with $\lambda = \beta = \infty$. In other words, there are no comparison noises. Moreover, as before, the two actions $\overline{A}_n^{(0)}$ and $\overline{A}_n^{(1)}$ are i.i.d. sampled from a distribution $\mu$ over $\mathcal{A}$. With this, we understand how this sampling distribution $\mu$ affects the results.

For a finite dataset $\mathcal{D}_0$ of size $N$, let $\mathcal{U}_{\mathcal{D}_0} \subset \mathcal{A}$ be the set consisting of all unique actions occurring in $\mathcal{D}_0$. Then, the informative set $\mathcal{U}_{\mathcal{D}_0}$ can be constructed with two types of actions : (i) actions not appearing in $\mathcal{U}_{\mathcal{D}_0}$ (ii) actions occurring in $\mathcal{U}_{\mathcal{D}_0}$ that have not 'lost' the comparison with any another action. We begin by constructing an algorithm for this analysis.

### A.2.1 Developing the Algorithm

For a finite dataset $\mathcal{D}_0$ of size $N$, let $\mathcal{U}_N \subset \mathcal{A}$ be the set consisting of all unique actions occurring in $\mathcal{D}_0$. Then, $\mathcal{U}$ can be constructed with two types of actions : (i) actions not appearing in $\mathcal{U}_N$ (ii) actions occurring in $\mathcal{U}_N$ that have not 'lost' the comparison with any another action. For this, let $\mathcal{C}_i$ be the set of comparisons from $\mathcal{D}_0$ involving action $a_i$ i.e. $\mathcal{C}_i = \left\{ \left( \overline{A}_n^{(0)}, \overline{A}_n^{(1)}, Y_n \right) ; \overline{A}_n^{(0)} = a_i \text{ or } \overline{A}_n^{(1)} = a_i , n \in [N] \right\}$. Hence, construct $\mathcal{U} = (\mathcal{A} \backslash \mathcal{U}_N) \cup \mathcal{W}_{\mathcal{U}_N}$, where $\mathcal{W}_{\mathcal{U}_N} := \{ a_i \in \mathcal{U}_N ; Y_j = a_i \ \forall \left( \overline{A}_j^{(0)}, \overline{A}_j^{(1)}, Y_j \right) \in \mathcal{C}_i \}$.

Note here that the conditions mentioned above are tight conditions, which can be analyzed in the case of uniform action sampling distribution ($\mu = \texttt{Uniform}(\cdot)$). However, in the case of an arbitrary distribution such analysis is intractable. In this case, we then only consider the sufficient condition to obtain complete ordering of actions. The sufficient condition to determine the optimal action with high probability is to sample each pair of actions at least once i.e. sample each of $\binom{K}{2}$ pairs once.

### A.2.2 Finding the Optimal Action Given a Large Dataset

Here, we show that the construction procedure as described above in Part A.2.1 yields in finding the optimal action given a large dataset. More formally, we show that if the action sampling distribution $\mu$ is not degenerate i.e. $\lim_{N \to \infty} P(a \in \mathcal{U}_N) > 0 \ \forall \ a \in \mathcal{A}$, then as $N \to \infty$, $\mathcal{U} \to \{A^*\}$ almost surely.

To see this, if $\mu$ is not degenerate, then $\lim_{N \to \infty} P(a \in \mathcal{U}_N) > 0 \ \forall \ a \in \mathcal{A}$. Then, $\lim_{N \to \infty} (\mathcal{A} \setminus \mathcal{U}_N) = \varnothing$. In addition, as $N \to \infty$, for all possible pairs of actions $(a_i, a_j)$ with $a_i \neq a_j \in \mathcal{A}$, we will have $\overline{A}_n^{(0)} = a_i$ and $\overline{A}_n^{(1)} = a_j$ for some $\left( \overline{A}_n^{(0)}, \overline{A}_n^{(1)}, Y_n \right) \in \mathcal{D}_0$. Due to the construction of $\mathcal{W}_{\mathcal{U}_N}$, we will also have $\lim_{N \to \infty} \mathcal{W}_{\mathcal{U}_N} = \{A^\star\}$ almost surely. This implies $\lim_{N \to \infty} \mathcal{U} = \{A^\star\}$.

Now that we know that the construction procedure of $\mathcal{U}_N$ is principled, we wish to generalize the result for finite size of the offline dataset $\mathcal{D}_0$.

### A.2.3 General Sample Complexity Analysis

In this section, we present results for the following question : In general, how large $N$ need be to ensure that $P(|\mathcal{U}| = \wr(K))$, or even $P(|\mathcal{U}| = 1)$ with high probability? We aim to derive this result for an arbitrary action sampling distribution $\mu$, however we begin by analyzing the case of uniform distribution i.e. $\mu \sim \texttt{Uniform}(\cdot)$.

**General analysis of probability of picking all $n$ items in $N$ trials.**
Here, we describe the general theoretical framework to bound the probabilities of picking *all* of $n$ given items in $N$ independent trials. We first begin with a uniform distribution over each of these $n$ items and later generalize to an arbitrary distribution $\mu$. Note that we derive a general result for $n$ items, which in our case corresponds to actions ($n = K$) or action pairs ($n = \binom{K}{2}$).

- **Uniform distribution.** We have $n$ items which are equally likely to be selected, so we can invoke the Stirling numbers of the second kind (or Stirling partition number) to get a bound on this probability. Stirling numbers of the second kind give the number of ways to partition a set of $u$ objects into $v$ non-empty subsets and is denoted by $S(u, v)$. For notation, we have $n$ items to be selected, $N$ as the number of trials.

Now, let $S_i$ be all the outcomes in which an item $i$ is *not* selected. For each $i$, $|S_i| = (n-1)^N$ and there are $\binom{n}{1}$ choices for $i$. For each $j \neq i$, $|S_j \cap S_i| = (n-2)^N$ and there are $\binom{n}{2}$ choices for $(i,j)$. Continuing in this manner to count the number of outcomes missing at least 1 number, we get

$$\left| \bigcup_{i=1}^{n} S_i \right| = \sum_{i=1}^{n} |S_i| - \sum_{j<i} |S_j \cap S_i| + \sum_{k<j<i} |S_k \cap S_j \cap S_i| - \dots$$
$$= \binom{n}{1}(n-1)^N - \binom{n}{2}(n-2)^N + \binom{n}{3}(n-3)^N - \dots$$

Since there are a total of $n^N$ total outcomes, we get the number of desired outcomes in which *all* possible numbers are rolled, denoted by $\#_{\text{desired}}$ as

$$\#_{\text{desired}} = n^N - \binom{n}{1}(n-1)^N + \binom{n}{2}(n-2)^N - \binom{n}{3}(n-3)^N + \dots.$$

Thus, the probability $p_{n,N}$ of picking all $n$ items in $N$ trials is $\frac{\#_{\text{desired}}}{n^N}$. Hence,

$$p_{n,N} = 1 - \binom{n}{1}\left(1 - \frac{1}{n}\right)^N + \binom{n}{2}\left(1 - \frac{2}{n}\right)^N - \binom{n}{3}\left(1 - \frac{3}{n}\right)^N + \dots$$
$$\Rightarrow p_{n,N} = \sum_{i=0}^{n}(-1)^i \binom{n}{i}\left(1 - \frac{i}{n}\right)^N \tag{13}$$

- **Arbitrary Distribution.** Assume now that the actions are sampled from an action sampling distribution $\mu$. Since we are forming action pairs for comparison, denote with $\mathring{\mu}_k$, the probability of sampling action pair $k := (i,j) \in [\binom{K}{2}]$ and with $\sum_k \mathring{\mu}_k = 1$, with $(i,j)$ representing the action pair $(a_i, a_j)$. Furthermore, this means that assume that $\mu_{\min}^2 \leq \mathring{\mu}_k \leq \mu_{\max}^2 \ \forall \ k$ for some arbitrary $0 < \mu_{\min} \leq \mu_{\max} < 1$.

  For this problem, let $T_i$ denote the random number of trials needed to sample item $i$ for the first time. The total number of trials needed can be then denoted by the random variable $T = \max(T_1, \dots, T_n)$. Note that $T_i$ is a geometric random variable with parameter $\mathring{\mu}_i$ because each new item obtained is of type $i$ with probability $\mathring{\mu}_i$, but now these variables are no more independent. Since the minimum of $T_i$ and $T_j$ is the number of trials needed to obtain either item $i$ or item $j$, it follows that for $j \neq i$, $\min(N_i, N_j)$ is a geometric random variable with parameter $\mathring{\mu}_i + \mathring{\mu}_j$ and the same holds true for the minimum of any finite number of these random variables. Hence, we can write,

$\mathbb{E}[T] = \mathbb{E}[\max_i T_i]$
$$= \sum_i \mathbb{E}[T_i] - \sum_{i<j} \mathbb{E}[\min(T_i, T_j)] + \sum_{i<j<k} \mathbb{E}[\min(T_i, T_j, T_k)] - \dots + (-1)^{n+1}\mathbb{E}[\min(T_1, \dots, T_n)]$$
$$= \sum_i \frac{1}{\mathring{\mu}_i} - \sum_{i<j} \frac{1}{\mathring{\mu}_i + \mathring{\mu}_j} + \sum_{i<j<k} \frac{1}{\mathring{\mu}_i + \mathring{\mu}_j + \mathring{\mu}_k} - \dots + (-1)^{n+1}\frac{1}{\mathring{\mu}_1 + \dots + \mathring{\mu}_n}$$

Recall that $\int_0^\infty e^{-tx}dx = \frac{1}{t}$. We also know the identity

$$1 - \prod_{i=1}^{n}(1 - e^{-t_i x}) = \sum_i e^{-t_i x} - \sum_{i<j} e^{-(t_i+t_j)x} + \dots + (-1)^{n+1}e^{-(t_1+\dots+t_n)x} \tag{14}$$

Using the above identity, and integrating it, we get

$$\mathbb{E}[T] = \int_0^\infty \left(1 - \prod_{i=1}^{n}\left(1 - e^{-\mathring{\mu}_i x}\right)\right)dx \tag{15}$$

**Lemma A.2.** *Let the action set be $\mathcal{A} = \{a_0, \ldots, a_K\}$, with a sampling distribution $\mu$ such that $0 < \mu_{min} \leqslant \mu_k \leqslant \mu_{max} < 1 \;\; \forall\; k \in [K]$. For the case of $\beta, \lambda \to \infty$, with some given $\epsilon \in (0, 1)$, the minimum size of the offline dataset to ensure $\mathcal{U}_{\mathcal{D}_0} = \{A^\star\}$ and hence is $(1 - \epsilon)$-informative is given by*

$$Uniform\ \mu\ :\ N \geqslant \frac{K^2 \ln K}{\epsilon} \qquad ; \qquad Arbitrary\ \mu\ :\ N \gtrsim \frac{\ln K}{\mu_{min}^2 \epsilon} \tag{16}$$

*Proof.* We first begin by proving the case of uniform action sampling distribution, and then extend the results to an arbitrary distribution.

**Uniform Distribution.**

If $|\mathcal{U}| = 1$, then $|\mathcal{A} \backslash \mathcal{U}_N| = 1 \wedge |\mathcal{W}_{\mathcal{U}_N}| = 0$ OR $|\mathcal{A} \backslash \mathcal{U}_N| = 0 \wedge |\mathcal{W}_{\mathcal{U}_N}| = 1$.

**Case 1.** $|\mathcal{A} \backslash \mathcal{U}_N| = 1$ and $|\mathcal{W}_{\mathcal{U}_N}| = 0$.

For the former, we simply do not want to select the optimal action while making action pairs, and hence the probability is:

$$P(|\mathcal{A} \backslash \mathcal{U}_N| = 1) \geqslant \left(1 - \frac{2}{K}\right)^N$$

For the latter, we use Equation (13) with $n = \binom{K-1}{2}$ to get

$$P(|\mathcal{W}_{\mathcal{U}_N}| = 0) \geqslant \sum_{i=0}^{\binom{K-1}{2}} (-1)^i \binom{\binom{K-1}{2}}{i} \left(1 - \frac{i}{\binom{K-1}{2}}\right)^N.$$

**Case 2.** $|\mathcal{A} \backslash \mathcal{U}_N| = 0$ and $|\mathcal{W}_{\mathcal{U}_N}| = 1$.

In this case, if all pairs are sampled at least once, the event $\{|\mathcal{W}_{\mathcal{U}_N}| = 1\}$ is a sufficient condition for event $\{|\mathcal{A} \backslash \mathcal{U}_N| = 0\}$ to occur. Hence, we use Equation (13) with $n = \binom{K}{2}$ to get the probability as:

$$P(|\mathcal{W}_{\mathcal{U}_N}| = 1) \geqslant \sum_{i=0}^{\binom{K}{2}} (-1)^i \binom{\binom{K}{2}}{i} \left(1 - \frac{i}{\binom{K}{2}}\right)^N.$$

Putting it all together, we need

$$1 - \epsilon \leqslant P(|\mathcal{A} \backslash \mathcal{U}_N| = 1) \cdot P(|\mathcal{W}_{\mathcal{U}_N}| = 0) + P(|\mathcal{W}_{\mathcal{U}_N}| = 1)$$

However, the above form is intractable to solve for a closed form solution. Hence, we use the Stirling number approximation for factorials (i.e. $\ln(n!) \approx n \ln(n) - n$) and approximation of the Stirling number of second kind i.e. $S(u, v) \leqslant n \ln(n) - n \ln(\ln(n)) + n \ln(k)$, where $S(u, v) = \sum_{i=0}^{v} \frac{(-1)^{v-i} i^u}{(v-i)! i!}$. In addition, we also Stirling's approximation to the binomial as $\binom{a}{b} \approx \frac{a^b}{b!}$ for $a >> b$. Using these, the expression simplifies to $N \geqslant \frac{K^2 \ln K}{\epsilon}$.

**Arbitrary Distribution.**

Similar to the case of uniform distribution, we still need

$$1 - \epsilon \leqslant P(|\mathcal{A} \backslash \mathcal{U}_N| = 1) \cdot P(|\mathcal{W}_{\mathcal{U}_N}| = 0) + P(|\mathcal{W}_{\mathcal{U}_N}| = 1)$$

However, a closed form solution for the above does not exist for the case of multiple actions. Instead, we aim to derive the result based on the sufficient condition for obtaining the optimal action : if all pairs of actions are sampled at least once, we know the optimal action. For this, we just need $P(|\mathcal{W}_{\mathcal{U}_N}| = 1) \geq 1 - \epsilon$.

Recalling from the analysis given above in Section A.2.3, $T_i$ denotes the random number of trials needed to sample item $i$ for the first time. The total number of trials needed can be then denoted by the random variable $T = \max(T_1, \ldots, T_n)$.

Now, since $T$ is a random variable denoting the total number of trials needed to obtain all $n$ items *at least* once, it can also be viewed as the *stopping time* for when the agent has collected all items. Hence, we are interested in the probability $P(T \leq N)$ i.e. the probability that this stopping time $T$ is less than the dataset size $N$. This is because the event $\{T \leq N\}$ is the event that by time (or dataset size) $N$, the agent has sampled all $n$ items.

We then also have $P(T \leq N) = 1 - P(T > N)$. Since $N$ is non-negative, we can bound the $P(T > N)$ using a concentration inequality as below using Equation (15).

$$P(T > N) \leq \frac{\mathbb{E}[T]}{N} = \frac{1}{N} \int_0^\infty \left( 1 - \prod_{i=1}^n \left( 1 - e^{-\mathring{\mu}_i x} \right) \right) dx$$

$$\Rightarrow P(T \leq N) \geq 1 - \frac{1}{N} \int_0^\infty \left( 1 - \prod_{i=1}^n \left( 1 - e^{-\mathring{\mu}_i x} \right) \right) dx \tag{17}$$

$$\geq 1 - \frac{1}{N} \int_0^\infty \left( \sum_i e^{-\mathring{\mu}_i x} - \sum_{i<j} e^{-(\mathring{\mu}_i + \mathring{\mu}_j)x} + \cdots + (-1)^{n+1} e^{-(\mathring{\mu}_1 + \cdots + \mathring{\mu}_n)x} \right) dx$$

$$\text{(using Identity (14))}$$

$$\geq 1 - \frac{1}{N\mu_{\min}^2} \left( \frac{\binom{n}{1}}{1} - \frac{\binom{n}{2}}{2} + \cdots + (-1)^{n+1} \frac{\binom{n}{n}}{n} \right) \tag{18}$$

$$\geq 1 - \frac{H_n}{N\mu_{\min}^2} \tag{19}$$

, where $H_n$ is the Harmonic sum of the first $n$ natural numbers. Now, we wish that $P(T \leq N) \geq 1 - \epsilon$. Using the bound above, we find that we need

$$N \geq \frac{H_n}{\mu_{\min}^2 \epsilon} \gtrsim \frac{\ln K}{\mu_{\min}^2 \epsilon}.$$

$$\square$$

## A.3 Understanding Multiple Actions and Finite Deliberateness

We can break the expected number of samples needed to find an optimal action into parts and then use a generalized version of the Coupon Collection problem, solution of which is known (Newman, 1960). The first deals with using Equation (11) to find the minimum samples needed to determine the more likely optimal action between two actions (one pair) with high probability of $(1 - \frac{\epsilon}{2n})$, where $n$ is total number of items. Here, $n$ would be the number of pairs i.e. $n = \binom{K}{2}$. The second part deals with finding the bound on total number of samples needed to determine the more likely optimal action for *every* such pair.

**Finding the better action in the $i^{\text{th}}$ item (pair).** The expected number of samples needed to find the better action can be calculated using Equation (11). Call this number $k_i$. So,

$$k_i \geq \frac{\ln\left( \left( \frac{2\binom{K}{2}}{\epsilon} - 1 \right) \left( \frac{1}{\Phi(x_i)} - 1 \right) \right)}{\beta \langle a_i^{(0)} - a_i^{(1)}, \theta_0 \rangle},$$

where $x_i := \dfrac{(a_i^{(0)} - a_i^{(1)})^T \mu_0}{\sqrt{\left(a_i^{(0)} - a_i^{(1)}\right)^T \Sigma_0 \left(a_i^{(0)} - a_i^{(1)}\right)}}$, $\Phi(\cdot)$ is the CDF of the standard Normal distribution, and $(a_i^{(0)}, a_i^{(1)})$ are the actions of the $i^{\text{th}}$ pair.

**Theorem 4.2.** *Let the action set $\mathcal{A}$ have size $K$ with a sampling distribution $\mu$ such that $0 < \mu_{min} \leqslant \mu_k \leqslant \mu_{max} < 1$, $\forall k \in [K]$. Given some $\epsilon \in (0,1)$ and finite $\beta < \infty$, let $\lambda \to \infty$. Then, the singleton set $\mathcal{U}_{\mathcal{D}_0} = \{A^\star\}$ is $(1 - \epsilon)$-informative if*

$$N > N_0 := \frac{\ln K + (k_{max} - 1) \ln \ln K}{\mu_{min}^2 \epsilon}, \quad where \tag{5}$$

$$k_{max} = \max_{i,j \in [K]} \frac{\ln\left(\left(\frac{2K^2}{\epsilon} - 1\right)\left(\frac{1}{\Phi(x_{i,j})} - 1\right)\right)}{\beta \langle a_i - a_j, \theta_0 \rangle}, \, and \, x_{i,j} = \frac{(a_i - a_j)^T \mu_0}{\sqrt{\left(a_i - a_j\right)^T \Sigma_0 \left(a_i - a_j\right)}},$$

*and, $N$ is the size of the preference dataset and $\Phi(\cdot)$ is the standard Normal CDF.*

*Proof.* Our sample complexity analysis to achieve constant Bayesian regret can be broken down into three main building blocks:

- Appendix A.1 and Lemma A.1 : there are only two actions ($|\mathcal{A}| = 2$) but we have finite deliberateness ($\beta < \infty$).

- Appendix A.2 and Lemma A.2: there are many actions ($|\mathcal{A}| = K$) but we have very high deliberateness ($\beta \to \infty$).

- Appendix A.3 : there are many actions ($|\mathcal{A}| = K$) and finite deliberateness, where we combine the results from the first two cases. In this case, we can break the expected number of samples needed to find an optimal action into two parts. The first deals with using Lemma A.1 to find the minimum samples needed to determine the more likely optimal action between two actions (one pair) with high probability of $(1 - \frac{\epsilon}{2n})$, where $n$ is total number of items. Here, $n$ would be the number of pairs i.e. $n = \binom{K}{2}$. The second part deals with finding the bound on total number of samples needed to determine the more likely optimal action for *every* such pair.

With this in mind, we prove the result below.

Newman (1960) gave a generalization of the coupon collector's problem when $m$ copies of each coupon need to be collected with total coupons being $n$. Let $T_m$ be the first time $m$ copies of *each* coupon are collected. We then know that $\mathbb{E}[T_m] = n \ln n + (m - 1)n \ln(\ln n)$.

Using similar analysis as before for a general action pair sampling distribution $\mu$ with $\mu_{\min} \leqslant \mu_i \leqslant \mu_{\max}$ for some $\mu_{\min}, \mu_{\max} \in [0,1)$ for all items (pairs) $i \in [n]$, we can derive the general sampling result. Adapting it to our setting, we need $k_i$ samples for $i^{\text{th}}$ pair, and we have $n = \binom{K}{2}$ pairs. Letting $T_{k_{\max}}$ denote the total number of samples needed to obtain $k_{\max}$ number of samples for each item (pair),

$$\mathbb{E}[T_{k_{\max}}] \leqslant \frac{1}{\mu_{\min}^2} \left[2 \ln(n) + (k_{\max} - 1) \ln(\ln(n))\right] \quad ; \quad n = \binom{K}{2} , \; k_{\max} = \max_{i \in [n]} k_i$$

Denoting $T_{k_{\max}}$ as the random stopping time when at least $k_{\max}$ occurrences of all $n$ items have been collected, we need $P(T_{k_{\max}} > N) \leqslant \frac{\epsilon}{2}$, where $N$ is the size of the offline dataset $\mathcal{D}_0$. Hence, using Markov inequality, we can bound it as:

$$N \geqslant \frac{\ln K + (k_{\max-1}) \ln \ln K}{\mu_{\min}^2 \epsilon} \quad where, \tag{20}$$

$$k_{\max} = \max_{i,j \in [K]} \frac{\ln\left(\left(\frac{2K^2}{\epsilon} - 1\right)\left(\frac{1}{\Phi(x_{i,j})} - 1\right)\right)}{\beta \langle a_i - a_j, \theta_0 \rangle} , \; x_{i,j} = \frac{(a_i - a_j)^T \mu_0}{\sqrt{\left(a_i - a_j\right)^T \Sigma_0 \left(a_i - a_j\right)}}$$

$\square$

### A.4 Regret Analysis Continued

In this appendix section, we provide the building block proofs that allow us to construct a prior-dependent Bayesian regret bound on the warmPref − PS algorithm. The heart of these proofs lies in constructing a $(1 - \epsilon)$-informative set $\mathcal{U}_{\mathcal{D}_0}$ from the offline dataset $\mathcal{D}_0$.

**Lemma A.3.** $\mathcal{U}_{\mathcal{D}_0}$ is $(1 - f_1)-$informative.

*Proof.* We construct $\mathcal{U}_{\mathcal{D}_0}$ as a *set* of actions that have been preferred to *at least* once in the offline dataset $\mathcal{D}_0$ and of actions that do not appear in the $\mathcal{D}_0$. Thus, $\mathcal{U}_{\mathcal{D}_0}$ contains at most $K$ actions.

Now, we consider the formulation below. Recall that $\overline{A}_n^{(0)}$ and $\overline{A}_n^{(1)}$ are i.i.d. sampled from the action set and each datapoint in the dataset $\mathcal{D}_0^i$, conditioned on $\vartheta, \beta$, is independent of $\mathcal{D}_0^j$ for $i \neq j$. Now,

$$P\left(A^\star \notin \mathcal{U}_{\mathcal{D}_0}\right) \leqslant P(A^\star \text{ has lost all comparisons in } \mathcal{D}_0) + P(A^\star \text{ is not present in } \mathcal{D}_0)$$

$$\leqslant \mathbb{E}\left[ \prod_{n=1}^{N} \frac{\exp\left(\beta \langle a_n, \vartheta \rangle\right)}{\exp\left(\beta \langle a_n, \vartheta \rangle\right) + \exp\left(\beta \langle A^\star, \vartheta \rangle\right)} + (1 - \mu_{\min})^{2N} \right]$$

$$\leqslant \mathbb{E}\left[ \prod_{n=1}^{N} \left( 1 - \frac{\exp\left(\beta \langle A^\star, \vartheta \rangle\right)}{\exp\left(\beta \langle a_n, \vartheta \rangle\right) + \exp\left(\beta \langle A^\star, \vartheta \rangle\right)} \right) \right] + (1 - \mu_{\min})^{2N} \qquad (21)$$

$$\leqslant \mathbb{E}\left[ \prod_{n=1}^{N} \left( 1 - \underbrace{\frac{1}{1 + \exp\left(-\beta \langle A^\star - a_n, \vartheta \rangle\right)}}_{\clubsuit} \right) \right] + (1 - \mu_{\min})^{2N}$$

, where $A^\star$ is a function of $\theta$ and thus a random variable as well. Looking closely at the term $\clubsuit$ above, it can be written as $P(Y_n = A^\star \mid \vartheta)$. We now analyze this term.

$$P(Y_n = A^\star \mid \vartheta) = \frac{1}{1 + \exp\left(-\beta \langle A^\star - a_n, \vartheta \rangle\right)}$$

$$= \left( 1 + \exp\left(\beta \langle A^\star - a_n, \theta - \vartheta \rangle - \beta \langle A^\star - a_n, \theta \rangle\right) \right)^{-1}$$

$$\geqslant \left( 1 + \exp\left(\beta \|A^\star - a_n\|_1 \|\theta - \vartheta\|_\infty - \beta \langle A^\star - a_n, \theta \rangle\right) \right)^{-1} \qquad \text{(Hölder's inequality)}$$

$$\geqslant \left( 1 + \exp\left(\beta \|\vartheta - \theta\|_\infty - \beta \langle A^\star - a_n, \theta \rangle\right) \right)^{-1} \qquad (\|A^\star - a_n\|_1 \leqslant 1 \ \forall \ a_n \in \mathcal{A})$$

Since $\vartheta - \theta \sim N(0, \mathbf{I}_d / \lambda^2)$, using the Dvoretzky–Kiefer–Wolfowitz inequality bound (Massart, 1990; Vershynin, 2010) implies

$$P\left(\|\vartheta - \theta\|_\infty \geqslant t\right) \leqslant 2d^{1/2} \exp\left(-\frac{t^2 \lambda^2}{2}\right).$$

Set $t = \sqrt{2 \ln(2d^{1/2} T)}/\lambda$ and define an event $\mathcal{E}_1 := \{\|\vartheta - \theta\|_\infty \leqslant \sqrt{2 \ln(2d^{1/2} T)}/\lambda\}$ such that $P(\mathcal{E}_1^c) \leqslant 1/T$. We decompose Equation (21) using Union Bound as:

$$P\left(A^\star \notin \mathcal{U}_{\mathcal{D}_0}\right) \leqslant \mathbb{E}\left[ \prod_{n=1}^{N} \left( 1 - P\left(Y_n = A^\star \mid \theta, \vartheta\right) \right) \mathbb{I}_{\mathcal{E}_1} \right] + P(\mathcal{E}_1^c) + (1 - \mu_{\min})^{2N}$$

$$\leqslant \mathbb{E}\left[ \prod_{n=1}^{N} \left( 1 - \left( 1 + \exp\left(\frac{\beta \sqrt{2 \ln(2d^{1/2} T)}}{\lambda}\right) \underbrace{\exp\left(-\beta \langle A^\star - a_n, \theta \rangle\right)}_{\blacktriangle} \right)^{-1} \right) \right] + \frac{1}{T} + (1 - \mu_{\min})^{2N}. \qquad (22)$$

Now, we define another event $\mathcal{E}_{(n)} := \{\langle A^\star - a_n, \theta \rangle \leqslant \Delta\}$. Based on $\mathcal{E}_{(n)}$ we analyze the $\blacktriangle$ term as follows.

$$\exp\left(-\beta\langle A^\star - a_n, \theta\rangle\right) = \mathbb{E}\big[\exp\left(-\beta\langle A^\star - a_n, \theta\rangle\right)\mathbb{I}_{\mathcal{E}_{(n)}}\big] + \mathbb{E}\big[\exp\left(-\beta\langle A^\star - a_n, \theta\rangle\right)\mathbb{I}_{\mathcal{E}_{(n)}^c}\big]$$
$$\leqslant \exp\left(0\right)P(\mathcal{E}_{(n)}) + \exp\left(-\beta\Delta\right)P(\mathcal{E}_{(n)}^c)$$
$$\leqslant P(\mathcal{E}_{(n)}) + (1 - P(\mathcal{E}_{(n)}))\exp\left(-\beta\Delta\right)$$

Plugging this back in Equation (22) we get,

$$P\left(A^\star \notin \mathcal{U}_{\mathcal{D}_0}\right) \leqslant \mathbb{E}\left[\prod_{n=1}^{N}\left(1 - \left(1 + \exp\left(\frac{\beta\sqrt{2\ln(2d^{1/2}T)}}{\lambda}\right)\left(\mathbb{I}_{\mathcal{E}_{(n)}} + (1 - \mathbb{I}_{\mathcal{E}_{(n)}})\exp\left(-\beta\Delta\right)\right)\right)^{-1}\right)\right] + \frac{1}{T} + (1 - \mu_{\min})^{2N} \tag{23}$$

Note that the random variable $\mathbb{I}_{\mathcal{E}_{(n)}}$ depends on the action sampling distribution $\mu$. Denote the probability of sampling this action $a_n$ by $\mu_n$, and as before we have $\mu$ supported by $[\mu_{\min}, \mu_{\max}]$. We first analyze this for any arbitrary $n \in [N]$ and study the the distribution of $\mathbb{I}_{\mathcal{E}_{(n)}}$ conditionaled on $A^\star$. Without loss of generality, we first condition on $A^\star = \mathring{a}$ for some $\mathring{a} \in \mathcal{A}$. For that, let $\rho(\cdot)$ be the univariate Gaussian distribution and $\theta_a = \langle a, \theta\rangle$ for any action $a$.

$$P\left(\mathbb{I}_{\mathcal{E}_{(n)}} = 1 \,|\, A^\star = \mathring{a}\right) = P\left(\mathbb{I}\left(\langle A^\star - a_n, \theta\rangle \leqslant \Delta\right) = 1 \,\big|\, A^\star = \mathring{a}\right)$$
$$= \frac{1}{P\left(A^\star = \mathring{a}\right)}P\left(\mathbb{I}\left(\langle A^\star - a_n, \theta\rangle \leqslant \Delta\right) = 1, A^\star = \mathring{a}\right)$$
$$= \frac{1}{P\left(A^\star = \mathring{a}\right)}P\left(\mathbb{I}\left(\theta_{a_n} \geqslant \theta_{\mathring{a}} - \Delta\right) = 1, \bigcap_{a \in \mathcal{A}}\{\theta_{\mathring{a}} \geqslant \theta_a\}\right)$$
$$= \frac{1}{P\left(A^\star = \mathring{a}\right)}\int_{\mathbb{R}}\left[\int_{\theta_{\mathring{a}} - \Delta}^{\infty}d\rho(\theta)\right]d\rho(\theta_{\mathring{a}})$$
$$= \frac{1}{P\left(A^\star = \mathring{a}\right)}\int_{\mathbb{R}}\left[\int_{\theta_{\mathring{a}} - \Delta}^{\theta_{\mathring{a}}}d\rho(\theta)\right]d\rho(\theta_{\mathring{a}}) \qquad (\text{since }\theta_a \leqslant \theta_{A^\star} \ \forall\, a) \tag{24}$$

Noticing that the term inside the integral can be represented as a distribution, we first find a normalizing constant to represent the probabilities. So, define

$$\Phi(\theta_{\mathring{a}}) = \int_{-\infty}^{\theta_{\mathring{a}}}(2\pi)^{-1/2}\exp(-x^2/2)\,dx \qquad ; \qquad g(\theta_{\mathring{a}}) = \frac{1}{\Phi(\theta_{\mathring{a}})}\int_{\theta_{\mathring{a}} - \Delta}^{\theta_{\mathring{a}}}d\rho(\theta)\,.$$

For fixed $\theta_{\mathring{a}}$, let $X_{\theta_{\mathring{a}}} \sim \text{Bernoulli}(1, g(\theta_{\mathring{a}}))$. With Eq. (24) and letting $d\mu(\theta_{\mathring{a}}) = \frac{\Phi(\theta_{\mathring{a}})}{P(A^\star = a_{\mathring{a}})}d\rho(\theta_{\mathring{a}})$ we have,

$$P\left(\mathbb{I}_{\mathcal{E}_{(n)}} = 1 \,|\, A^\star = \mathring{a}\right) = \int_{\mathbb{R}}P(X_{\theta_{\mathring{a}}} = 1)\frac{\Phi(\theta_{\mathring{a}})}{P(A^\star = \mathring{a})}d\rho(\theta_{\mathring{a}}) = \int_{\mathbb{R}}P(X_{\theta_{\mathring{a}}} = 1)\,d\mu(\theta_{\mathring{a}})\,. \tag{25}$$

Plugging this back in Equation (23) and upper bounding the probabilities we get,

$$P\left(A^\star \notin \mathcal{U}_{\mathcal{D}_0}\right) \leqslant \sum_{a \in \mathcal{A}}\int_{\mathbb{R}}P(X_{\theta_a} = 0)d\mu(\theta_a)P\left(A^\star = a\right)\left(1 - \left(1 + \exp\left(\beta\left(\lambda^{-1}\sqrt{2\ln(2d^{1/2}T)} - \Delta\right)\right)\right)^{-1}\right)^{N} \cdot \mu_{\max}^{N}$$
$$+ \frac{1}{T} + (1 - \mu_{\min})^{2N}$$
$$\leqslant \sum_{a \in \mathcal{A}}\int_{\mathbb{R}}P(X_{\theta_a} = 0)\left(1 - \left(1 + \exp\left(\beta\left(\lambda^{-1}\sqrt{2\ln(2d^{1/2}T)} - \Delta\right)\right)\right)^{-1}\right)^{N} \cdot \mu_{\max}^{N}\,d\mu(\theta_a)P\left(A^\star = a\right)$$
$$+ \frac{1}{T} + (1 - \mu_{\min})^{2N}$$
$$\leqslant \int_{\mathbb{R}}\mathbb{E}_{\mathring{a} \in \mathcal{A}}\left[\left(1 - \left(1 + \exp\left(\beta\left(\lambda^{-1}\sqrt{2\ln(2d^{1/2}T)} - (1 - X_{\theta_{\mathring{a}}})\Delta\right)\right)\right)^{-1}\right)^{N}\right] \cdot \mu_{\max}^{N}d\mu(\theta_{\mathring{a}}) + \frac{1}{T} + (1 - \mu_{\min})^{2N}, \tag{26}$$

where $\mu_{\max}$ is used to obtain the exponent $N$ by accounting for the sampling distribution $\mu$, and last step follows from the uniformity of each action being optimal. Finally, we need to find the supremum of $g(\theta_{\mathring{a}})$ and hence Equation (26). Recall that,

$$g(\theta_{\mathring{a}}) = \frac{1}{\Phi(\theta_{\mathring{a}})} \int_{\theta_{\mathring{a}}-\Delta}^{\theta_{\mathring{a}}} d\rho(\theta) = \frac{\int_{\theta_{\mathring{a}}-\Delta}^{\theta_{\mathring{a}}} d\rho(\theta)}{\int_{-\infty}^{\theta_{\mathring{a}}} d\rho(\theta)} = \frac{\int_{-\infty}^{\theta_{\mathring{a}}} d\rho(\theta) - \int_{-\infty}^{\theta_{\mathring{a}}-\Delta} d\rho(\theta)}{\int_{-\infty}^{\theta_{\mathring{a}}} d\rho(\theta)} = 1 - h_\Delta(\mathring{a})$$

, where $h_\Delta(\mathring{a}) := \frac{\int_{-\infty}^{\theta_{\mathring{a}}-\Delta} d\rho(\theta)}{\int_{-\infty}^{\theta_{\mathring{a}}} d\rho(\theta)}$. Setting $\nabla_{\mathring{a}} h_\Delta(\mathring{a}) = 0$ and analyzing $\nabla_{\mathring{a}}^2 h_\Delta(\mathring{a}) > 0$, we find that

$$g(\theta_{\mathring{a}}) \leqslant 1 - \Delta \exp\left(-\frac{(2\theta_{\mathring{a}} - \Delta)\Delta}{2}\right) \leqslant \min(1, \Delta) . \tag{27}$$

Finally we, decompose Equation (26) based on the event $\mathcal{E}_2 := \{X_{\theta_{\mathring{a}}} = 0\}$, and upper bound the probability to simplify. Setting $\Delta = \ln(T\beta)/\beta$, we conclude with the following bound:

$$P\left(A^\star \notin \mathcal{U}_{\mathcal{D}_0}\right) \leqslant \left(1 - \left(1 + \exp\left(\beta\left(\lambda^{-1}\sqrt{2\ln(2d^{1/2}T)} - (K-1)\min(1, \ln(T\beta)/\beta)\right)\right)\right)^{-1}\right)^N + \frac{1}{T} + (1 - \mu_{\min})^{2N} \tag{28}$$

$\square$

**Lemma A.4.** $\mathbb{E}[|\mathcal{U}_{\mathcal{D}_0}|] \leqslant f_2$.

*Proof.* Recall that $\mathcal{U}_{\mathcal{D}_0}$ is a *set* of actions that have been preferred to *at least* once in the offline dataset $\mathcal{D}_0$ and of actions that do not appear in the $\mathcal{D}_0$. We first see that $\mathbb{E}[|\mathcal{U}_{\mathcal{D}_0}|] = \sum_{k=1}^K k \cdot P\left(|\mathcal{U}_{\mathcal{D}_0}| = k\right)$. Define an event $\mathcal{E}_a = \{\langle A^\star - a, \theta \rangle \leqslant \Delta\}$ and analyze as follows,

$$\begin{aligned}
\mathbb{E}[|\mathcal{U}_{\mathcal{D}_0}|] &= \mathbb{E}\left[\sum_{a \in \mathcal{A}} \mathbb{I}(a \in \mathcal{U}_{\mathcal{D}_0})\right] = \sum_{a \in \mathcal{A}} \mathbb{E}\left[\mathbb{I}(a \in \mathcal{U}_{\mathcal{D}_0})\mathbb{I}(\mathcal{E}_a) + \mathbb{I}(a \in \mathcal{U}_{\mathcal{D}_0})\mathbb{I}(\mathcal{E}_a^c)\right] \\
&\leqslant K \min(1, \Delta^2/2) + \frac{1}{T} + \mathbb{E}\left[\sum_{a \in \mathcal{A}} \mathbb{I}\left(a \in \mathcal{U}_{\mathcal{D}_0}\right)\mathbb{I}(\mathcal{E}_a^c)\right],
\end{aligned} \tag{29}$$

where the second step follows from the event $\mathcal{E}_a$ and analysis done before : break down the indicator variable conditioning on arbitrary $A^\star = \mathring{a} \in \mathcal{A}$, and use Poisson approximation to bound the probability. Now, analyze term in expectation above.

$$\mathbb{E}\left[\sum_{a\in\mathcal{A}}\mathbb{I}(a\in\mathcal{U}_{\mathcal{D}_0})\mathbb{I}(\mathcal{E}_a^c)\right] = \mathbb{E}\left[\sum_{n=1}^{N}\sum_{a\in\mathcal{A}}P\left(Y_n = a, \langle A^\star - a, \theta - \vartheta\rangle + \langle A^\star - a, \vartheta\rangle \geqslant \Delta \mid \theta, \vartheta\right)\mathbb{I}(\mathcal{E}_a^c)\right]$$

$$\leqslant \sum_{n=1}^{N}\mathbb{E}\left[\sum_{a\in\mathcal{A}}P\left(Y_n = a, \langle A^\star - a, \vartheta\rangle \geqslant \Delta - \sqrt{2\ln(2d^{1/2}T)}/\lambda \mid \vartheta\right)\mathbb{I}(\mathcal{E}_a^c)\right]$$

$$\leqslant N\mathbb{E}\left[\sum_{a,b\in\mathcal{A}\,;\,\langle A^\star - a,\vartheta\rangle\geqslant\Delta-\sqrt{2\ln(2d^{1/2}T)}/\lambda}(1 + \exp\left(\beta\langle b - a, \vartheta\rangle\right))^{-N}\right] \tag{30}$$

$$\leqslant N\mathbb{E}\left[\sum_{a\in\mathcal{A}\,;\,\langle A^\star - a,\vartheta\rangle\geqslant\Delta-\sqrt{2\ln(2d^{1/2}T)}/\lambda}(1 + \exp\left(-\beta\langle A^\star - a, \vartheta\rangle\right))^{-N}\right]$$

$$\leqslant \frac{N(K-1)}{T\beta}\left(1 + \exp\left(-\beta\left(\lambda^{-1}\sqrt{2\ln(2d^{1/2}T)} + (K-1)\min(1,\Delta)\right)\right)\right)^{-N}$$

$$\leqslant \frac{NK}{T\beta}\left(1 + \exp\left(-\beta\left(\lambda^{-1}\sqrt{2\ln(2d^{1/2}T)} + (K-1)\min(1,\Delta)\right)\right)\right)^{-N}$$

Putting all of this together, we obtain the bound below.

$$\mathbb{E}[|\mathcal{U}_{\mathcal{D}_0}|] \leqslant K\min(1,\Delta^2/2) + \frac{NK}{T\beta}\left(1 + \exp\left(-\beta\left(\lambda^{-1}\sqrt{2\ln(2d^{1/2}T)} + (K-1)\min(1,\Delta)\right)\right)\right)^{-N} + \frac{1}{T}. \tag{31}$$

Of course, $|\mathcal{U}_{\mathcal{D}_0}|$ cannot exceed $K$, so we have with the choice of $\Delta = \ln(T\beta)/\beta$,

$$\mathbb{E}[|\mathcal{U}_{\mathcal{D}_0}|] \leqslant \min\left(K\min\left(1,\frac{\ln^2(T\beta)}{2\beta^2}\right) + \frac{NK}{T\beta}\left(1 + \exp\left(-\beta\lambda^{-1}\sqrt{2\ln(2d^{1/2}T)} + (K-1)\min(1,\ln(T\beta)/\beta)\right)\right)^{-N} + \frac{1}{T}, K\right). \tag{32}$$

$\square$

**Lemma A.5.** *If $\mu_{min} > 0$, then set $\mathcal{U}_{\mathcal{D}_0}$ constructed above is $(1 - f_1)$-informative, and $\mathbb{E}[|\mathcal{U}_{\mathcal{D}_0}|] \leqslant f_2$.*

*Proof.* Combining Lemma A.3 and Lemma A.4 we have the desired result. $\square$

### A.5 Constructing Surrogate Loss Function

This section contains proofs of construction of the surrogate loss function as described in Section 5.

**Lemma A.6.** *At time $t$, the MAP estimate of $(\theta, \vartheta)$ can be constructed by solving the following equivalent optimization problem:*

$$(\theta_{opt}, \vartheta_{opt}) = \operatorname*{argmax}_{\theta,\vartheta} P(\theta, \vartheta \mid \mathcal{D}_{t-1})$$

$$\equiv \operatorname*{argmin}_{\theta,\vartheta} \mathcal{L}_1(\theta, \vartheta) + \mathcal{L}_2(\theta, \vartheta) + \mathcal{L}_3(\theta, \vartheta),$$

$$where, \qquad \mathcal{L}_1(\theta, \vartheta) := \frac{1}{2}\sum_{s=1}^{t-1}\left(R_s - \langle A_s, \theta\rangle\right)^2, \tag{33}$$

$$\mathcal{L}_2(\theta, \vartheta) := -\sum_{n=1}^{N}\beta\langle \overline{A}_n^{(Y_n)}, \vartheta\rangle + \ln\left(e^{\beta\langle\overline{A}_n^{(0)},\vartheta\rangle} + e^{\beta\langle\overline{A}_n^{(1)},\vartheta\rangle}\right),$$

$$\mathcal{L}_3(\theta, \vartheta) := \frac{\lambda^2}{2}||\theta - \vartheta||_2^2 + \frac{1}{2}(\theta - \mu_0)^T\Sigma_0^{-1}(\theta - \mu_0).$$

*Proof.* We first analyze the posterior distribution of $\vartheta, \theta$ given the offline dataset $\mathcal{D}_0$, optimize it by treating these random variables as parameters.

$$
\begin{aligned}
\operatorname*{argmax}_{\theta, \vartheta} P(\theta, \vartheta \,|\, \mathcal{D}_{t-1}) &= \operatorname*{argmax}_{\theta, \vartheta} P(\mathcal{D}_{t-1} \,|\, \theta, \vartheta) \cdot P(\theta, \vartheta) \\
&= \operatorname*{argmax}_{\theta, \vartheta} \ \ln P(\mathcal{D}_{t-1} \,|\, \theta, \vartheta) + \ln P(\theta, \vartheta) \\
&= \operatorname*{argmax}_{\theta, \vartheta} \underbrace{\ln P(\mathcal{H}_{t-1} \,|\, \mathcal{D}_0, \theta, \vartheta)}_{\mathcal{L}_1} + \underbrace{\ln P(\mathcal{D}_0 \,|\, \theta, \vartheta)}_{\mathcal{L}_2} + \underbrace{\ln P(\theta, \vartheta)}_{\mathcal{L}_3}
\end{aligned}
\tag{34}
$$

Then,

$$
\begin{aligned}
\mathcal{L}_1 &= \sum_{s=1}^{t-1} \underbrace{\ln P(A_s \,|\, \mathcal{D}_{s-1}, \theta, \vartheta)}_{\text{indep. of } \theta, \vartheta \ \Rightarrow \ \text{constant}} + \ln P(R_s \,|\, A_s, \theta, \vartheta) \\
&= \text{constant} \ - \ \frac{t-1}{2} \ln\left(\frac{2\pi}{\sigma^2}\right) - \frac{1}{2} \sum_{s=1}^{t-1} \left(R_s - \langle A_s, \theta \rangle\right)^2. \\
\mathcal{L}_2 &= \sum_{n=1}^{N} \ln\left(\left(\overline{A}_n^{(0)}, \overline{A}_n^{(1)}, Y_n\right) \,|\, \theta, \vartheta\right) \\
&= \sum_{n=1}^{N} \ln\left(Y_n \,|\, \overline{A}_n^{(0)}, \overline{A}_n^{(1)}, \theta, \vartheta\right) + \underbrace{\ln P\left(\overline{A}_n^{(0)}, \overline{A}_n^{(1)} \,|\, \theta, \vartheta\right)}_{\text{indep. of } \theta, \vartheta \ ; \ \text{depends on } \mu \ \Rightarrow \ \text{constant}} \\
&= \sum_{n=1}^{N} \beta \langle \overline{A}_n^{(Y_n)}, \vartheta \rangle - \ln\left(e^{\beta \langle \overline{A}_n^{(0)}, \vartheta \rangle} + e^{\beta \langle \overline{A}_n^{(1)}, \vartheta \rangle}\right) + \text{constant} \\
\mathcal{L}_3 &= \ln P(\vartheta \,|\, \theta) + \ln P(\theta) \\
&= \frac{d}{2} \ln\left(\frac{2\pi}{\lambda^2}\right) - \frac{\lambda^2}{2} \|\theta - \vartheta\|_2^2 - \frac{1}{2} \ln\left(|2\pi \Sigma_0|\right) - \frac{1}{2}(\theta - \mu_0)^T \Sigma_0^{-1}(\theta - \mu_0).
\end{aligned}
\tag{35}
$$

Hence, final surrogate loss function is

$$
\begin{aligned}
\mathcal{L}(\theta, \vartheta) &= \mathcal{L}_1(\theta, \vartheta) + \mathcal{L}_2(\theta, \vartheta) + \mathcal{L}_3(\theta, \vartheta), \qquad \text{where} \\
\mathcal{L}_1(\theta, \vartheta) &= \frac{1}{2} \sum_{s=1}^{t-1} \left(R_s - \langle A_s, \theta \rangle\right)^2 \\
\mathcal{L}_2(\theta, \vartheta) &= -\sum_{n=1}^{N} \beta \langle \overline{A}_n^{(Y_n)}, \vartheta \rangle + \ln\left(e^{\beta \langle \overline{A}_n^{(0)}, \vartheta \rangle} + e^{\beta \langle \overline{A}_n^{(1)}, \vartheta \rangle}\right) \\
\mathcal{L}_3(\theta, \vartheta) &= \frac{\lambda^2}{2} \|\theta - \vartheta\|_2^2 + \frac{1}{2}(\theta - \mu_0)^T \Sigma_0^{-1}(\theta - \mu_0).
\end{aligned}
\tag{36}
$$

Finally the problem in Equation (34) becomes equivalent as follows:

$$
(\theta_{opt}, \vartheta_{opt}) = \operatorname*{argmax}_{\theta, \vartheta} P(\theta, \vartheta \,|\, \mathcal{D}_t) \equiv \operatorname*{argmin}_{\theta, \vartheta} \mathcal{L}(\theta, \vartheta)
\tag{37}
$$

$\square$

### A.6  warmPref-PS with Online Feedback (`warmTSOF`)

Here, we present an extension to warmPref $-$ PS, where the agent has the option to ask for feedback during the online phase.

---

**Algorithm 3** warm Thompson Sampling with Preference Feedback (`warmTSOF`)

---

1: **Input:** Horizon $T$, offline dataset $\mathcal{D}_0$, set of arms $\mathcal{A}$, knowledgeability $\lambda$, deliberateness $\beta$, feedback cost $c$.
2: **for** $t = 1, 2, \ldots, T$ **do**
3:     Sample a set of perturbations $\mathcal{P}_t = \{\zeta_s, \omega_n, \theta', \vartheta'\}$.
4:     Solve Equation (9) using this set $\mathcal{P}_t$ to find $(\widehat{\theta}_t, \widehat{\vartheta}_t)$.
5:     Let $A_t^1, A_t^2$ be s.t. $\langle A_t^1, \widehat{\theta}_t \rangle \geqslant \langle A_t^2, \widehat{\theta}_t \rangle \geqslant \langle A, \widehat{\theta}_t \rangle \ \forall \ A \in \mathcal{A} \backslash \{A_t^1, A_t^2\}$.
6:     Compute $\epsilon_t = \texttt{get\_epsilon}(c, \mathcal{D}_t, t, \lambda, \beta)$.
7:     **if** $\left| \langle A_t^1, \widehat{\theta}_t \rangle - \langle A_t^2, \widehat{\theta}_t \rangle \right| < \epsilon_t$ **then**
8:         Ask for feedback on $(A_t^1, A_t^2)$ and receive $Y_t \in \{0, 1\}$.
9:         Update $\mathcal{D}_t \leftarrow \mathcal{D}_t \cup \{A_t^1, A_t^2, Y_t\}$.
10:        Update posterior using Equation (9) to get new $(\widetilde{\theta}_t, \widetilde{\vartheta}_t)$.
11:        Set $A_t = \text{argmax}_{a \in \mathcal{A}} \langle a, \widetilde{\theta}_t \rangle$ and $c_t = c$.
12:     **else**
13:        Set $A_t = A_t^1$ and $c_t = 0$.
14:     **end if**
15:     Take action $A_t$ to receive reward $R_t - c_t$.
16:     Set $\mathcal{D}_{t+1} = \mathcal{D}_t \cup \{A_t, R_t\}$.
17: **end for**

---

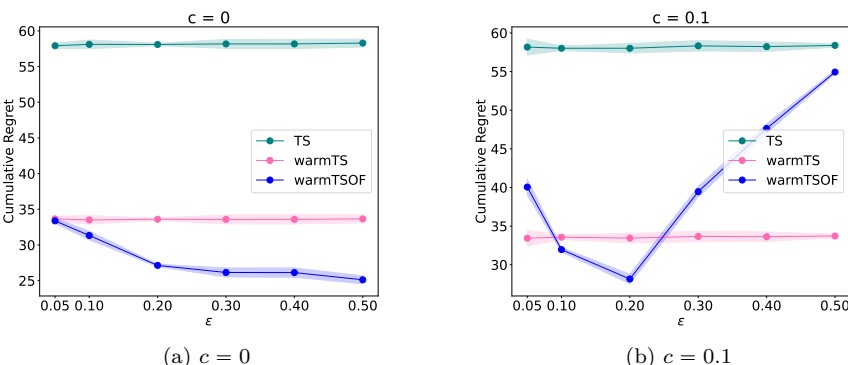

Figure 4: Performance of `warmTSOF` with varying cost of feedback $c$.

**Problem Formulation.** We present the warm Posterior Sampling with Online Feedback (`warmTSOF`) algorithm wherein the learning agent can ask for preference feedback between two actions at some cost. We present some preliminary empirical results and defer the theoretical analysis to future work. This has applications in active learning (Ren et al., 2021; Margatina et al., 2021) and crowd-sourcing data from experts for large language models (Mishra et al., 2021; Korbak et al., 2023).

Consider, now in addition to Algorithm 2 (`warmTS`), the agent at any time, has the option to ask for *online* preference feedback between two actions. For simplicity, we assume the rater for this feedback is the same rater who generated the offline dataset $\mathcal{D}_0$.

Let the cost incurred for this feedback on actions $A_t^1$ and $A_t^2$ be $c_t = c \in \mathbb{R}$ if agent asks for feedback, else $c_t = 0$. So, this feedback takes the form $\{A_t^1, A_t^2, Y_t\}$, and $Y_t \in \{0, 1\}$ and the reward the agent receives then becomes $R_t - c_t$.

The agent should incorporate the expected current rewards for all actions, cost of feedback, and expert competency into the decision making process. The core idea is to only initiate feedback retrieval process if top-two expected rewards of all actions are 'close'. This idea finds its in beginnings in the Top-Two Thompson

Sampling procedure ([Russo](#), [2016](#)). See `warmTSOF` (Algorithm [3](#)) for exact details. The `get_epsilon(·)` function will be decided through analysis.

**Performance.** See Figure [4](#) for performance comparison. Experiments are run with size of offline dataset $N = 20$, deliberateness $\beta = 10$, and knowledgeability $\lambda = 10$. In addition, we let number of arms $k = 10$, dimension of environment $d = 4$, and horizon $T = 300$, all averaged over 100 runs (random seeds). For baselines, we consider the traditional and warm Thompson Sampling (`TS` and `warmTS`).

## A.7 Evaluation using `DPO` and `IPO`

DPO ([Rafailov et al.](#), [2024](#)) is an alternative approach to the RL paradigm, which avoids the training of a reward model altogether. The loss that DPO optimizes to obtain the optimal policy, given an empirical dataset $\mathcal{D} = \{y_w, y_l\}$ of the winning (preferred) $y_w$ and losing (not preferred) $y_l$ outputs (arms in our bandit setting), as a function of the reference policy $\pi_{\text{ref}}$ and regularization strength $\tau \in \mathbb{R}_+$, is given by:

$$\pi^\star_{\text{DPO}} = \operatorname*{argmin}_\pi \quad \mathbb{E}_{(y_w, y_l) \sim \mathcal{D}} \left[ -\log \sigma \left( \tau \log \left( \frac{\pi(y_w)}{\pi(y_l)} \right) - \tau \log \left( \frac{\pi_{\text{ref}}(y_w)}{\pi_{\text{ref}}(y_l)} \right) \right) \right]$$

, where $\sigma(\cdot)$ denotes the sigmoid function.

IPO is an instance of the $\Psi$PO algorithm ([Gheshlaghi Azar et al.](#), [2024](#)) . The loss function that IPO optimizes is given by,

$$\pi^\star_{\text{IPO}} = \operatorname*{argmin}_\pi \quad \mathbb{E}_{(y_w, y_l) \sim \mathcal{D}} \left[ h_\pi(y_w, y_l) - \frac{1}{2\tau} \right]^2 \text{ where}, \ h_\pi(y, y') := \log \left( \frac{\pi(y) \pi_{\text{ref}}(y')}{\pi(y') \pi_{\text{ref}}(y)} \right) .$$

Note here that DPO and IPO are purely offline learning algorithms that only work with preference datasets. DPO and IPO cannot be trivially extended to our problem setting i.e. fixed offline preference dataset with active online numerical reward learning. Comparing DPO and IPO based solely on offline datasets is not fair. Hence, for a more fair comparison, we consider an offline-online variant of DPO with $\epsilon$-greedy online exploration. The pseudo code is as follows:

- **Input:** offline preference data $\mathcal{D}_0$, $\epsilon$, $\min_a r^\star(a)$, DPO parameters
- **Offline learning:** use DPO to learn a policy $\pi$, use $\pi$ to infer a reward model $r$ such that $\min_{a \in \mathcal{A}} r(a) = \min_{a \in \mathcal{A}} r^\star(a)$, where $r^\star(\cdot) : \mathbb{R}^d \to \mathbb{R}$ is true reward model.
- **Online learning:** at each time $t = 1, \ldots, T$
  - with probability $\epsilon$, choose $A_t$ uniformly randomly; otherwise, choose $A_t = \operatorname{argmax}_a r(a)$.
  - observe reward from the environment, which is $r^\star(A_t)$ plus noise.
  - update the reward model $r$ based on the received reward.

For training, mini-batches are drawn uniformly with replacement from $\mathcal{D}_0$ and optimized using `DPO` and `IPO` for 20k steps. Policy is encoded simply as $\pi_\psi(a_i) = \operatorname{softmax}(\psi)_i$ for an action $a_i \in \mathcal{A}$ using a vector $\psi \in \mathbb{R}^K$, and is optimized for 20k steps using Adam ([Kingma & Ba](#), [2014](#)) with a learning rate of 0.015 and mini-batch size 12. Reference policy is $\pi_{\text{ref}}$ is chosen to be uniform over the action space, regularization is set at 0.1, and $\epsilon = 0.16$ for best results.

## A.8 Additional Experiments

We conducted additional experiments to validate warmPref-PS in the finite armed linear bandit setting. We first begin with experiments on the MovieLens dataset ([Harper & Konstan](#), [2015](#)), followed by various settings of the quality of the dataset.

To conduct empirical evaluation within the finite-armed linear bandit framework, we additionally construct an offline preference dataset using the MovieLens ratings corpus (Harper & Konstan, 2015). MovieLens provides explicit numerical ratings of the form $(u, i, r_{u,i})$, where $u$ indexes a user, $i$ indexes an item (movie), and $r_{u,i} \in \{1, \ldots, 5\}$ is an absolute score. Since our problem formulation does not include a user or contextual component, and since our offline dataset consists solely of pairwise preferences of the form $\mathcal{D}_0 = \{(A_n^{(0)}, A_n^{(1)}, Y_n)\}_{n \in [N]}$, we preprocess MovieLens so that it serves only as a source of *item-level* reward structure and then generate preferences according to the Bradley-Terry model in Equation (1).

**Constructing arm features.** We treat each movie as an arm $A \in \mathcal{A} \subset \mathbb{R}^d$. To obtain a fixed $d$-dimensional feature vector for each arm, we fit a standard matrix factorization model to the MovieLens rating matrix and use the resulting item embedding as the feature vector. Concretely, we learn latent factors $(p_u, q_i)$ via $r_{u,i} \approx p_u^\top q_i$, and set the arm feature for movie $i$ to be $A_i := q_i \in \mathbb{R}^d$. This yields a consistent linear representation of the action set that integrates naturally into the linear bandit model.

**Constructing the reward model used for preference generation.** Since our online bandit environment is defined by a latent parameter $\theta \in \mathbb{R}^d$, we estimate a "ground-truth" reward parameter $\theta^\star$ from the MovieLens data by solving a ridge regression problem

$$\theta^\star := \operatorname*{argmin}_{\theta \in \mathbb{R}^d} \sum_i \left(\bar{r}_i - A_i^\top \theta\right)^2 + \lambda_{\text{reg}} \|\theta\|_2^2,$$

where $\bar{r}_i$ is the empirical mean rating of movie $i$. The resulting reward model, parameterized by $\theta^\star$, specifies a consistent reward structure over arms, which is used *only to synthesize preference responses* and is never revealed to the learner. Note that there are other ways to learn a reward model (e.g.: in the RLHF pipeline Ouyang et al. (2022)), but for our setting the above representation suffices.

**Synthesizing the offline preference dataset.** Having fixed movies $\mathcal{A}$ and $\theta^\star$, we sample action pairs $(A_n^{(0)}, A_n^{(1)})$ i.i.d. from a distribution $\mu$ over $\mathcal{A}$, as assumed in Section 2. For each sampled pair, we generate a preference label $Y_n$ using Equation (1). This produces an offline dataset whose statistical properties match exactly the structure assumed in our theoretical development. Importantly, MovieLens ratings are *not* directly converted into binary preferences; rather, they serve only to estimate a reward vector $\theta^\star$ from which Bradley–Terry preferences are generated. Thus, the offline dataset conforms precisely to the preference-generation model used throughout the paper.

**Generating online rewards.** In the online phase, since we do not have access to a simulator, for action (movie) $A_t = \text{movie}_i \in \mathcal{A}$ taken at round $t \in [T]$, we let the noisy reward be $R_t = \bar{r}_i + \epsilon$, where $\epsilon \sim \mathcal{N}(0, \sigma^2)$ for some $\sigma^2 \in \mathbb{R}$, where $\bar{r}_i$ is again the empirical mean rating of $\text{movie}_i$ calculated from the dataset. The remaining steps in the algorithm are same as Algorithm 1, and estimation of $\beta$ is done according to the discussion in Section 6.

**Evaluation Protocol.** For experiments we choose the top $K = 200$ rated movies (number of ratings, not the ratings themselves) as the action set, dimension $d = 10$, $\lambda = 10$, dataset size $N = 10^3$, and horizon $T = 300$. We averaged over 5 runs (with random seeds). For easy interpretation, we let $\mu \sim \text{Unif}(\cdot)$. Results are presented in Figure 5, where we observe higher variance in cumulative regret for all algorithms as compared to the synthetic dataset experiments in Section 6. Nevertheless, warmPref-PS, when warmstarted with an offline dataset and allowed for minimal online learning, still outperforms the baselines.

### A.9  Ablation study (cont.)

**Effect of number of arms $K$ and online rounds $T$.**  Here, we study how cumulative regret scales with $K$ and $T$. For evaluation, we let $d = 8, \lambda = 100, \beta = 8$, and $N = 25$. Empirical cumulative regret values are averaged over 5 runs with independent seeds. See Figure 6 for results.

**Effect of Action Space Dynamics.**  We next study how the dynamics of the action space affect cumulative regret. Specifically, how (i) the relationship between action pairs measured by their correlation ($\rho$), and (ii) the dimensionality of the environment vector $\theta \in \mathbb{R}^d$, affect cumulative regret. Table 1 shows that the

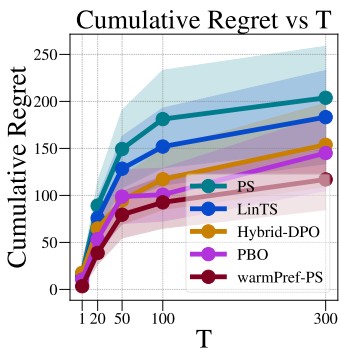

Figure 5: Cumulative regret on the MovieLens dataset.

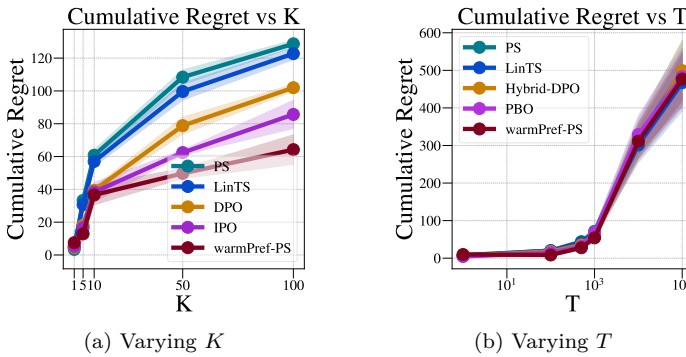

(a) Varying $K$         (b) Varying $T$

Figure 6: Cumulative regret with varying $K$ and $T$.

performance of all these posterior sampling methods degrades as dimensionality of the environment and correlation between action increases. However, warmPref − PS still outperforms the baselines and enjoys a lesser performance degradation than PS as $d$ and $\rho$ increase.

Table 1: Effect of dimensionality and correlation within the action space on cumulative regret.

|  | PS | LinTS | warmPref − PS |
|---|---|---|---|
| $d = 2, \rho = 0.1$ | $58.21 \pm 0.45$ | $53.23 \pm 0.64$ | $\mathbf{32.65 \pm 1.78}$ |
| $d = 2, \rho = 0.8$ | $61.36 \pm 1.23$ | $56.32 \pm 0.97$ | $\mathbf{33.98 \pm 3.07}$ |
| $d = 5, \rho = 0.1$ | $60.42 \pm 0.82$ | $55.71 \pm 0.41$ | $\mathbf{34.12 \pm 3.05}$ |
| $d = 5, \rho = 0.8$ | $64.21 \pm 1.57$ | $59.55 \pm 1.35$ | $\mathbf{34.77 \pm 2.94}$ |

