# OpenReview forum: "Online Bandit Learning with Offline Preference Data"
_TMLR — Rejected by TMLR_

### Review · Reviewer_RxYJ · 2025-11-06

**Summary Of Contributions:**

This work focuses on bandit learning and investigates a warm-up phase utilizing offline preference data before initiating the standard reinforcement learning process with an absolute reward. The authors account for imperfect users generating the preference data and propose a posterior sampling algorithm (WarmPref-PS). They provide a theoretical guarantee for the regret bound under mild assumptions, and simulation results support the efficiency of the proposed method.

**Audience:**

Yes

**Audience Explanation:**

In reinforcement learning, collecting online reward feedback is costly, while warming up with offline data can reduce this cost. Although the original algorithm is computationally inefficient, researchers have proposed a practical approximation method for warmPref-PS, which can be used in simulation experiments.

**Claims And Evidence:**

Yes

**Claims Explanation:**

For the proposed algorithm, the authors provide theoretical guarantees and rigorous proofs of regret bound, and simulation results also support the effectiveness of the proposed method.

**Requested Changes:**

Concerns Regarding Theoretical Results and Experiments

1. The selection of candidate set $\mathcal{U}_0$ is too trivial: "it contains all actions that have been preferred to another action $c$ in the offline dataset $D_0$ and also includes any actions that do not appear in the dataset."

Based on this definition, when the offline dataset is increasing, the set $\mathcal{U}_0$ will first decrease from the full set to a subset (when some actions appear and are dominated by other actions) and finally recover the full set (when the dataset is large enough, even a worse action may at least once beat a better action). Therefore, the regret guarantee in Theorem 4.4 will first decrease and then revert to the result without offline data. This highly contradicts the motivation, as we should expect that with a larger offline dataset, the performance will continue to improve and converge to a limit.

It seems more reasonable to have a careful design for the candidate set $\mathcal{U}_0$, for example, one focused on the ratio that one action is preferred to another, not just requiring preference only once.

2. It is not clear what the benefit of WarmPref-PS is. It seems all improvement comes from the high-probability candidate set $\mathcal{U}_0$. If we already know the optimal action falls in the candidate set with high probability, then the problem is reduced to a standard bandit problem with $|\mathcal{U}_0|$ arms, and any standard multi-arm bandit algorithm can achieve similar performance.

3. The author proposed a practical approximation method for the simulation. However, if we want to test the practical performance rather than only the theoretical guarantee, it is better to have some results for real-world data rather than just simulation.

---

> ### Author Response · Authors · 2025-11-10
>
> We wish to thank you for a careful review of the paper, and we try to address your concerns below:
>
> 1. Regarding $\mathcal{U}\_{\mathcal{D}\_{0}}$ and offline dataset: '' … when the offline dataset is increasing … '': please note that the offline dataset is given and fixed, and is independent of the learner’s online actions. In addition, we do not make any assumptions on how the dataset is generated - the actions selected for rater preference labeling might already be 'good’ actions, or they might be 'bad’ actions that need labeling (the 'good’ actions are then held out and do not appear in the dataset), or the action sampling distribution is such that the 'good’ actions have low sampling probability than 'bad’ actions. Without making additional assumptions, it is theoretically impossible to devise a construction criteria for  $\mathcal{U}\_{\mathcal{D}\_{0}}$ that is optimal for the varied 'quality’ of $\mathcal{D}\_{0}$. The criteria we have used in this paper is one of many (as also mentioned in the footnote of Page 6), and one that integrates well within our framework and allows for sound theoretical analysis. In addition, from Theorem 4.4 we do indeed have better performance with increasing dataset size i.e. $\tilde{f}\_{1}$ decreases exponentially with $N$.
>
>
> 2. Please note that the keyword in “If we already know the optimal action falls in the candidate set …” is **if**. Given a fixed offline dataset it is not trivial to construct a $\mathcal{U}\_{\mathcal{D}\_0}$ such that the optimal action lies in $\mathcal{U}\_{\mathcal{D}_\{0}}$. The procedure to do so along with its theoretical justification is the first contribution. Secondly, none of the standard multi-arm bandit algorithms (the baselines in our paper for example) can integrate preference feedback with reward feedback, and hence cannot exploit the information in the given offline dataset. This can be seen in the empirical results as well, where warmPref-PS achieves $25\\% \sim 55\\%$ cumulative regret reduction as compared to baselines, which can be attributed to warmPref-PS weighting the information from the offline dataset with the competence of the rater who generated it.
>
>
> 3. Thank you for this comment. While at the moment we could not find any large-scale *opensourced* datasets available for experiments, please let us know of any resources that may help us strengthen the empirical results.
>
> Please let us know if you have any more questions and we would be happy to discuss. We hope that we were able to address your concerns.

---

### Review · Reviewer_rw2t · 2025-11-16

**Summary Of Contributions:**

In this paper, the authors consider a Bayesian bandit problem with a finite number of arms and stochastic rewards, in which the learner is also provided with additional data before the start of the game for offline optimisation.

The novelty of this paper resides in the fact that this extra information is given in the form of preference feedback between pairs of actions. Crucially, these rankings are fallible as they are made to represent manual labeling. In this model, these preferences follow a noisy Bradley-Terry model, where they use the parameter $\beta$ to specify how deliberate the rating is ($\beta = 0$ is a random guess and $\beta \to \infty$ prefers the arm with the highest reward). Another parameter $\lambda$ characterizes how well the rater knows the environment (a large $\beta$ but low $\lambda$ would represent a rater that is very certain of their decisions but has a flawed representation.
The combination of these two parameters allows for a pretty detailed representation of the behaviors of the raters.

Using this extra information, the authors develop the warmPref-PS algorithm, which is a standard Posterior sampling approach that is known for its great performance for Bayesian bandit problems. The key contribution of this algorithm is the construction of a set of arms that has both a small size and is informative about the best arm with high probability.
This step of conversion of the preference information into that set allows the authors to use a result from Hao et al. (2023) (from the reference list of the paper) to bound the Bayesian regret of the algorithm when the offline data provides an informative subset of arms.
Then, the authors derive a Bayesian regret bound for the performance of the algorithm.

Concerning implementation, the authors first address the problem of intractability and then provide a wide set of experiments. Notably, they highlight the fact that their algorithm outperforms all baselines even when the preference information is not very informative (with varying amounts of preference data, certainty of the rater, and knowledge of the environment by the raters).

The authors also hypothesize that their approach would work for the problem of bandits with infinitely many arms.

**Additional Comments:**

Overall, the paper is interesting and the framework is novel.

**Audience:**

Yes

**Audience Explanation:**

The learning theory community is part of the audience of TMLR, and they will be interested in this paper. Bandit algorithms are a cornerstone of LT, and the novelty of using preference ranking for an offline warm-up should be of interest.

**Broader Impact Concerns:**

no concerns

**Claims And Evidence:**

Yes

**Claims Explanation:**

The main results proposed in the paper build upon a rich and well-studied literature in Bayesian bandits. The proofs are well-detailed in the appendix, and they seem correct to the best of my knowledge.

I would point out that there are a couple of statements that lack clarity in the paper and that need to be addressed.
Notably, the last sentence of section 4 seems to be lacking a claim: the fact that the approximate loss function works in the infinitely many armed case is fine, but as far as I can tell, no result was given to show that it "performs substantially better than other available baselines". This sentence is either imprecise, if this statement is related to the finite-arm case, or without justification if it is indeed about the infinitely many arms.

There is also a lack of clarity on the parameters that the algorithm needs to know ahead of time to optimize the algorithm and obtain the guarantees in the theorem.

**Requested Changes:**

The paper is well-written, but I expect that this approach could be of interest outside of the Bayesian bandit community. Adding some clarification about the Bayesian bandit framework could be helpful (though I recognize that it is not the responsibility of these authors to introduce the Bayesian bandit framework).


Concerning the changes, the first description of the construction of $D_0$ in section 4.1 was slightly confusing to read at first and would benefit from further clarification.
In Theorem 4.6, it would be really important to clarify which parameters are given to the algorithm ahead of time. As it stands, it is also very unclear whether the result holds no matter whether the preference feedback was helpful. The effect of the quality of the preference feedback and the raters is also fairly obscure in this theorem, and the statement that we reach sublinear Bayesian regret only when $N \to \infty$ isn't very practical (and in the experiments the preference data is helping, so something should be said about how big $N$ has to be in this main section and not just in the appendix).

 I would also recommend changing the sentence that I mentioned above in the infinitely many arms case (at the end of section 4).

---

> ### Author Response · Authors · 2025-11-18
>
> Firstly, thank you for your time in writing a detailed review of the paper. Regarding your comments:
>
> 1. Regarding the last sentence of Section 4: thank you for pointing out the ambiguity. Here, we are referring to the finite-armed case (i.e. “warmPref- PS algorithm that works for the infinite armed setting as well and performs substantially better than available baselines *in the finite-armed setting*”). The sentence has been placed prematurely, and would be more appropriate in the experimental section of the paper, where we do indeed see that warmPref- PS outperforms baselines across a variety of settings. Thank you for pointing this out, and we will also add a brief discussion about the knowledge of parameters needed by the algorithm.
>
>
> 2. Regarding ‘Requested Changes’: We will also add a brief introduction for the Bayesian bandit framework, which might help reach a wider audience. Thank you for this comment. As for rest of the changes, we shall make further clarifications in the revised version of the paper, and also add a few comments on how the regret bound varies with changing competence ($\beta, \lambda \to 0$ or $\beta, \lambda \to \infty$) and dataset size.
>
> Thank you for helping improve the paper; please let us know if you have any further questions.

---

> > ### Comment · Reviewer_rw2t · 2025-11-24
> >
> > Dear authors,
> >
> > Thank you for the answers and the proposed modifications. I believe that the other reviewers raised interesting points and that the difficulty of building a meaningful set $\mathcal U$.
> >
> > I would also note that one of the reviewers also asked about the convergence rate of $\tilde f_1$ in the main theorem. As it stands, when $N$ is large, you state that the second part of the bound drops to 0, but for low values of $N$ that term dominates the regret. Can you add a discussion of how large $N$ needs to be for the second part of the theorem not to dominate?
> >
> > Thank you/

---

> > > ### Author Response · Authors · 2025-11-27
> > >
> > > Thank you for this suggestion, and we too believe it will help improve readability and intuition of the result. We will make the addition after other reviewers have responded, thank you

---

### Review · Reviewer_LsHh · 2025-11-19

**Summary Of Contributions:**

This paper introduces a new framework for online bandit learning augmented with offline preference data from a possibly imperfect expert. The authors model the expert’s deliberateness (β) and knowledgeability (λ) and assume that preferences follow a noisy Bradley–Terry model around the true linear reward parameter. Before online learning begins, the algorithm uses these offline preferences to construct an informed prior over the reward parameter, leading to a Preference-Warmed Posterior Sampling (warmPref-PS) approach.

During the online phase, the learner interacts directly with the environment, receiving numeric reward feedback only (not new preferences). An optional extension (warmTSOF) allows querying new preferences at a cost, but this is secondary to the main method. The paper presents theoretical analyses on offline data informativeness and Bayesian regret, introduces a bootstrapped convex surrogate for tractable implementation, and validates the approach in synthetic experiments.

**Audience:**

Yes

**Audience Explanation:**

At least some individuals in TMLR’s audience—particularly those working on bandit learning, preference-based reinforcement learning, human-in-the-loop optimization, or RLHF—would be interested in this paper’s findings. The work contributes a conceptually novel bridge between offline preference data and online reward-based bandit learning, with solid theoretical grounding and clear relevance to mixed-feedback learning systems.

While the empirical evaluation is limited to synthetic settings, the theoretical insights (e.g., on data informativeness and regret reduction through preference-informed priors) are likely to appeal to TMLR readers interested in theoretical machine learning, online learning theory, and human feedback integration. Thus, even though broader empirical validation would strengthen it, the paper’s ideas and framework are of clear interest to a segment of the TMLR community.

**Broader Impact Concerns:**

No major ethical concerns, but a few aspects could merit discussion in a broader impact statement. The proposed method aims to combine human preference data with online learning, which is directly relevant to RLHF and human-in-the-loop AI systems. This connection introduces potential ethical implications around:
- If the offline preferences are collected from imperfect or unrepresentative human raters, the model may inherit or amplify those biases during online fine-tuning.
- The framework assumes access to offline human preference datasets, but the paper does not address data provenance, consent, or privacy considerations.
-  When used for systems that interact with humans (e.g., recommender systems or language models), competence modeling could be misused to weight human feedback unequally, raising fairness or accountability questions.

These issues are not critical for the current theoretical/synthetic study but would become important for real-world applications. A short Broader Impact section acknowledging these potential concerns and mitigation strategies (e.g., ensuring diverse raters, fairness-aware modeling, transparency in preference collection) would be helpful.

**Claims And Evidence:**

No

**Claims Explanation:**

The paper is technically strong, but the **empirical evaluation is quite limited** to fully assess the proposed setting and methods. Specifically:

* All experiments use *synthetic linear bandit* environments. The paper does not evaluate on any *standard preference-learning benchmarks* (e.g., dueling-bandit, MovieLens, or RLHF datasets), which weakens the claim of practical relevance.

* The evaluation compares warmPref-PS only with linear bandit baselines (PS, LinTS) and a simplified Hybrid-DPO. It omits comparisons to established *preference-based or dueling-bandit methods*, such as *DBGD (Dueling Bandit Gradient Descent)* and *PBO (Preferential Bayesian Optimization)*.

*  The study does not include a simple *“offline → reward → online”* pipeline—first fitting a reward model from preferences, then running LinTS. This baseline would help isolate the contribution of the joint Bayesian integration compared to standard pretraining.

*  The study does not explore the *extreme regime* where the offline posterior is so informative that online learning becomes unnecessary, nor the *complementary regime* where long online learning alone achieves near-optimality (i.e., offline data become redundant).

* The horizon (T) is relatively small (≈10³), so the experiments remain in a transient regime where offline data dominate. There is no demonstration that purely online learning eventually catches up to warmPref-PS as (T) increases.

* Both the theory and implementation assume a *linear reward model*. The framework cannot directly handle nonlinear reward mappings, and no nonlinear baselines are tested.

**Requested Changes:**

Suggestions for Improvement:
- Evaluate on at least one standard preference benchmark or RLHF-style dataset to demonstrate real-world relevance.
- Include preference-based baselines (e.g. DBGD, PB2O, BanditNet) and a two-step pretraining baseline to isolate gains from joint modeling.
- Discuss how the approach could extend to nonlinear reward models (e.g., kernelized or neural approximations).
- Clarify the scope of the online feedback (numeric only) in the main text.
- Discuss the case when reward feedback is noisy, or even in online phase one can only get preference feedback.
- Add long-horizon experiments to illustrate when offline data become redundant and pure online learning suffices.
- Consider extending the theoretical analysis toward posterior concentration or convergence guarantees.

---

> ### Author Response · Authors · 2025-11-21
>
> We wish to thank you for a careful and thorough review of the paper, and try to address your concerns below:
>
> 1. Regarding empirical evaluation: Thank you for providing the resources for additional datasets. In the revised version we will add experiments on the MovieLens dataset. Please note that DBGD is a purely online algorithm that cannot learn from *static* offline feedback (the setting of our paper). Hence, any comparison with warmPref-PS  is unfair even with a learned reward model. For PBO, it natively only learns from preference feedback (in both offline and online phases) and cannot incorporate reward feedback. Learning a reward model to provide preferences is still not appropriate for the online phase since PBO will receive information regarding *two* actions in the online phase, whereas in our setting we only get reward feedback for *one* action in the online phase. Nevertheless, we will add it to our baselines to strengthen the empirical results. Regarding the ‘extreme’ and ‘complementary’ regimes, please see Figure 2. The extreme regime corresponds to the bottom right graph (near-expert rater and large offline dataset size), and the complementary regime corresponds to the top left graph (extremely poor rater and small dataset size). In addition, please also see Figure 6 in Appendix A.9 where we see that long-horizon online learning yields comparative performance across baselines. Since our goal is to minimize this online learning (as discussed in Section 1), the benefit of warmPref-PS is more apparent when the algorithms are constrained to limited online rounds.
>
>
> 2. Regarding requested changes: Since the MovieLens dataset is an offline dataset, we would need to learn a proxy reward model from the dataset and simulate the online phase of all algorithms. Regarding preference feedback in the online phase, please see Appendix A.6 where we discuss this briefly, but it is currently beyond the scope of our paper. Regarding extending the analysis to function approximation settings and providing additional convergence guarantees, we thank you for this suggestion, but it is currently beyond the scope of our paper and we defer it to future work. Finally, we will include a short societal impact section in the paper to make the paper more relevant to practical settings.
>
> Thank you for your insightful comments and suggestions, and we hope that we were able to address your concerns. Please let us know if you have any more questions.

---

> ### Comment · Reviewer_LsHh · 2025-12-16
>
> I appreciate the authors' responses and the updates made to the paper. While the majority of my initial concerns have been addressed, several key points remain unclarified.
>
> Specifically, I still do not see an adequate response to my comments regarding:
> - The "offline $\rightarrow$ reward $\rightarrow$ online" pipeline baseline.
> - A discussion of the case where reward feedback is noisy, or when only preference feedback is available during the online phase.
>
> Furthermore, while the authors designate some concerns for "future work," I believe a deeper analysis is warranted now. For example, regarding the non-linear reward problem: if one can assume that the linear model can adequately approximate the non-linear reward (to a certain theoretical degree), can the authors provide a guarantee for the proposed approach?
>
> In summary, I believe the authors need to dedicate more effort to thoroughly addressing my comments.
>
> Finally, I must note that the current form of the responses is inappropriate and difficult to follow. The authors should provide a clear, point-by-point response to each concern and question raised.

---

### Author Response · Authors · 2025-11-22
**Updated Manuscript**

To all the reviewers,

Thank you for taking the time to review the paper. We have tried to incorporate the changes you have mentioned in the reviews and uploaded an updated manuscript, please see. Changes are coloured in pink for easy parsing. In addition, experiments on MovieLens dataset are added in Appendix A.8. Please let us know if any questions or concerns. Thank you

---

### Decision · Action_Editor_yi9n · 2025-12-29

**Recommendation:** Reject

**Additional Comments:**

If the authors can fix the concern I discussed above, it can be a good paper for both TMLR journal and major conferences. However, I tend to be conservative about the correctness and hope the authors can make more clarification and justification to polish this manuscript to a better shape.

**Audience:**

Yes

**Audience Explanation:**

We are lacking some theoretical understanding on how to leverage the preference-based data and this manuscript provides an initial trail, though not perfect.

**Claims And Evidence:**

No

**Claims Explanation:**

The authors formulate the problem of online contextual bandit with additional offline preference data and proposed a corresponding posterior sampling algorithm for solving it. Most of the claim are well supported by theoretical and empirical claim, but there are still some concerns from reviewers and myself. The most important one comes from reviewer RxYJ, which states that as $N\to\infty$, $|\mathcal{U}_{\mathcal{D}_0}| \to K$, which I think intuitively makes sense but contradicts Lemma 4.5. I have checked the proof of Lemma 4.5 in Lemma A.4. In Equation (30) I don't directly follow about the second inequality with the term $(1 + \exp(...))^{-N}$, as we already take each individual trails out with the $N$ outside the expectation and the probability inside should be the chance that $a$ is selected. Even there may be some chances that both reviewer RxYJ and I misunderstood the proof here, I think the authors should make this step more clear and interpret why we can have such counter-intuitive results. For the sake of correctness we should be more careful on this. Besides, I think Reviewer LsHh raises some reasonable concerns about the empirical evaluation. Given there are no benchmarks on this problem yet, it's understandable on this but with the concerns on the theoretical correctness, I think the authors can have a chance to both improve theoretical and empirical results.

**Resubmission Of Major Revision:**

The authors may consider submitting a major revision at a later time.